EMBO
Molecular Medicine

# Targeting miR-34a/*Pdgfra* interactions partially corrects alveologenesis in experimental bronchopulmonary dysplasia

Jordi Ruiz-Camp[1,2], Jennifer Quantius[2], Ettore Lignelli[1,2], Philipp F Arndt[2], Francesco Palumbo[1,2], Claudio Nardiello[1,2], David E Surate Solaligue[1,2], Elpidoforos Sakkas[1,2,†], Ivana Mižíková[1,2,‡], José Alberto Rodríguez-Castillo[1,2], István Vadász[2], William D Richardson[3], Katrin Ahlbrecht[1,2], Susanne Herold[2], Werner Seeger[1,2] & Rory E Morty[1,2,*] iD

## Abstract

**Bronchopulmonary dysplasia (BPD) is a common complication of preterm birth characterized by arrested lung alveolarization, which generates lungs that are incompetent for effective gas exchange. We report here deregulated expression of miR-34a in a hyperoxia-based mouse model of BPD, where miR-34a expression was markedly increased in platelet-derived growth factor receptor (PDGFR)α-expressing myofibroblasts, a cell type critical for proper lung alveolarization. Global deletion of miR-34a; and inducible, conditional deletion of miR-34a in PDGFRα[+] cells afforded partial protection to the developing lung against hyperoxia-induced perturbations to lung architecture. *Pdgfra* mRNA was identified as the relevant miR-34a target, and using a target site blocker *in vivo*, the miR-34a/*Pdgfra* interaction was validated as a causal actor in arrested lung development. An antimiR directed against miR-34a partially restored PDGFRα[+] myofibroblast abundance and improved lung alveolarization in newborn mice in an experimental BPD model. We present here the first identification of a pathology-relevant microRNA/mRNA target interaction in aberrant lung alveolarization and highlight the translational potential of targeting the miR-34a/*Pdgfra* interaction to manage arrested lung development associated with preterm birth.**

**Keywords** bronchopulmonary dysplasia; hyperoxia; lung development; miR-34a; platelet-derived growth factor
**Subject Category** Respiratory System

## Introduction

Bronchopulmonary dysplasia (BPD), a serious complication of preterm birth (Jobe, 2016), is characterized by arrested alveolarization of lungs of infants, arising from oxygen toxicity and mechanical injury during oxygen supplementation to manage respiratory failure. How these insults impair lung alveolarization is unclear (Surate Solaligue *et al*, 2017; Morty, 2018).

Lung development includes progressive subdivision of airspaces to expand alveoli number, thereby increasing gas-exchange surface area; and progressive thinning of septa to minimize gas diffusion distance (Pozarska *et al*, 2017). Alveolar myofibroblasts, which express αSMA, facilitate alveolarization (Vaccaro & Brody, 1978; Morrisey & Hogan, 2010; Hogan *et al*, 2014) by generating elastin cables that drive formation of secondary septa, which divide existing airspaces by squeezing the pre-existing alveoli with an elastin net, or pulling septal invaginations into airspaces (Branchfield *et al*, 2016). Myofibroblasts localize to alveolar entry rings during alveolarization (McGowan *et al*, 2008; Ntokou *et al*, 2015), exhibit phenotypic plasticity (Endale *et al*, 2017; McGowan & McCoy, 2017) and are marked by platelet-derived growth factor (PDGF) receptor (PDGFR)α, a mediator of normal (Boström *et al*, 1996, 2002; Gouveia *et al*, 2018) and aberrant (Oak *et al*, 2017) alveologenesis. Reduced levels of PDGFRα have also been noted in mesenchymal cells from human neonates that develop BPD (Popova *et al*, 2014).

How myofibroblast function is disturbed during aberrant alveolarization is not known, but a role for microRNA has been proposed, since deregulation of microRNA has been noted in clinical and experimental BPD (Nardiello & Morty, 2016), although no study has validated a causal role for any microRNA/mRNA interaction in alveolarization or BPD. We report here that the miR-34a/*Pdgfra*

---

1 Department of Lung Development and Remodelling, Max Planck Institute for Heart and Lung Research, Member of the German Center for Lung Research (DZL), Bad Nauheim, Germany
2 Department of Internal Medicine (Pulmonology), University of Giessen and Marburg Lung Center (UGMLC), Member of the German Center for Lung Research (DZL), Giessen, Germany
3 Wolfson Institute for Biomedical Research, University College London, London, UK
*Corresponding author. Tel: +49 6032 705 271; Fax: +49 6032 705 471; E-mail: rory.morty@mpi-bn.mpg.de
†Present address: Department of Clinical Genomics, SciLifeLab, Stockholm, Sweden
‡Present address: Regenerative Medicine Program, Sinclair Centre for Regenerative Medicine, Ottawa Hospital Research Institute, Ottawa, ON, Canada

interaction is disease relevant, and can be therapeutically targeted to partially restore lung alveolarization under pathological conditions. These data highlight a new mediator, and druggable target, in arrested alveolarization associated with preterm birth.

# Results and Discussion

### miR-34a is the most deregulated lung microRNA species in experimental BPD

BPD is modeled by exposure of newborn mice to hyperoxia (Nardiello *et al*, 2017a,b). Changes in microRNA expression during hyperoxia (85% $O_2$) exposure were detected by microarray (GEO accession number GSE89666). The steady-state levels of 10 and four microRNA species, respectively, were deregulated at post-natal day (P)5 and P14 (Fig 1A). These time-points represent the peak and near-completion phases, respectively, of bulk secondary septation in normally developing lungs (Morrisey & Hogan, 2010; Warburton *et al*, 2010). Levels of miR-34a-5p were the most consistently and appreciably increased of all microRNA species, implicating miR-34a-5p as a candidate mediator of arrested alveolarization. Independent validation by real-time RT–PCR revealed that miR-34a-5p levels were increased at P3, P5, and P14 in hyperoxia-exposed lungs (Fig 1B), with little or no impact on miR-34b-5p or miR-34c-5p (Fig 1B), or miR-34a-3p, miR-34b-3p, or miR-34c-3p (Fig 1C) levels noted. Levels of miR-34a-5p were consistently elevated over the P3-P14 hyperoxia-exposure time-course, in comparison with normoxia (21% $O_2$)-exposed lungs that exhibited normal alveolarization (Fig 1B). Together, these data highlight miR-34a-5p as a candidate mediator of arrested alveolarization.

### Global loss of miR-34a partially restores lung alveolarization in experimental BPD

Consistent with the arrested alveolarization that forms the hallmark of the BPD animal model, a 71% decrease in total alveoli number (Fig 2A and B; Appendix Table S1) and 10% increase in mean septal thickness (Fig 2A and C; Appendix Table S1) were noted in hyperoxia-exposed wild-type mouse lungs at P14, mimicking perturbations to lung structure noted in clinical BPD cases (Jobe, 2016; Nardiello *et al*, 2017b). Ablation of miR-34a (miR-34a$^{-/-}$ mice) partially protected against the impact of hyperoxia on alveolarization (Fig 2A; Appendix Table S1), with alveoli numbers increased by 47% (Fig 2B); and septal thickness decreased to even thinner than that noted in healthy mice (Fig 2C), compared to wild-type hyperoxia-exposed controls. No compensatory increase in miR-34b or miR-34c levels was noted in miR-34a$^{-/-}$ mice (Appendix Fig S1A). In contrast, dual ablation of miR-34b/miR-34c (miR-34bc$^{-/-}$ mice), without a compensatory increase in miR-34a levels (Appendix Fig S1B), did not impact alveoli number during hyperoxia-driven arrest of alveolarization (Fig 2D and E; Appendix Table S2). However, protection against hyperoxia-driven septal thickening in miR-34bc$^{-/-}$ mice was noted (Fig 2F), perhaps related to the increased levels of the 3p strands of miR-34b and miR-34c in the lungs of hyperoxia-exposed mice (Appendix Fig S1B). These data implicate miR-34a as mediator of arrested alveolarization associated with hyperoxia, an idea reinforced by detection of miR-34a expression with a *lacZ*-tagged miR-34a gene-trap in the septa of

developing lungs, with increased β-galactosidase staining evident after hyperoxia exposure (Fig 2G; Appendix Fig S2).

### miR-34a in PDGFRα$^+$ cells contributes to aberrant lung alveolarization

An *in silico* analysis identified two miR-34a-binding sites in the *Pdgfra* 3′-UTR (Fig 3A) (Silber *et al*, 2012; Garofalo *et al*, 2013). The PDGF-AA ligand and PDGFRα are key mediators of alveolarization (Boström *et al*, 1996, 2002), and reduced PDGFRα levels in mesenchymal cells are reported in human neonates that develop BPD (Popova *et al*, 2014). A synthetic miR-34a mimic reduced PDGFRα protein levels *in vitro* in MLg cells, a mouse lung fibroblast cell line, suggesting that a miR-34a/*Pdgfra* interaction occurs in mouse lung fibroblasts (Fig 3B), where increased miR-34 family microRNA transcripts (Fig 3C) and reduced *Pdgfra* mRNA transcripts (Appendix Fig S3) were noted in hyperoxia-exposed MLg cells. To explore this idea *in vivo*, exposure of newborn mice to hyperoxia (85% $O_2$) reduced lung PDGFRα protein levels at P5 (Fig 3D), which is the peak phase of bulk alveolarization (Morrisey & Hogan, 2010; Warburton *et al*, 2010). Treatment of MLg cells *in vitro* with antimiR-34a, which neutralizes miR-34a, partially protected steady-state PDGFRα protein levels against the impact of hyperoxia exposure, while an inert ("scrambled") antimiR did not (Fig 3E). These data support the contention that hyperoxia-driven elevations in miR-34a levels negatively regulated PDGFRα abundance. PDGFRα$^+$ cells were isolated from P5 mouse lungs by FACS (Appendix Fig S4A), where *in vivo* hyperoxia exposure had driven a dramatic increase in miR-34a levels in PDGFRα$^+$ cells (Fig 3F, Appendix Fig S5), accompanied by reduced *Pdgfra* (Appendix Fig S4B) and *Acta2* (Appendix Fig S4C) mRNA levels. The magnitude of the impact of hyperoxia on miR-34a levels in PDGFRα$^+$ cells was considerably larger than that observed in lung homogenates, highlighting the PDGFRα$^+$ cell as being particularly susceptible to hyperoxia-driven effects on miR-34a during alveologenesis.

To address miR-34a function in PDGFRα$^+$ cells, a mouse strain carrying a conditional, tamoxifen-inducible deletion of miR-34a in *Pdgfra*-expressing cells was generated (denoted miR-34a$^{i\Delta PC/i\Delta PC}$; Fig 3G) and was validated by demonstrating reduced miR-34a expression in PDGFRα$^+$ cells (Fig 3H). Ablation of miR-34a in PDGFRα$^+$ cells protected against hyperoxia-driven arrest of alveolarization (Fig 3I; Appendix Table S3), where approximately double the number of alveoli was noted in hyperoxia-exposed mice in which miR-34a expression was blocked in PDGFRα$^+$ cells (Fig 3J). Ablation of miR-34a expression in PDGFRα$^+$ cells did not impact hyperoxia-provoked perturbations to septal thickness (Fig 3K), which we attribute to the tamoxifen solvent, Miglyol, a complex fatty acid-derivative mixture, which we propose limited the impact of hyperoxia on septal thickening analogous to that reported for chemically related cottonseed oil (Nardiello *et al*, 2017b), since Miglyol alone is known to attenuate normal lung development (Fehl *et al*, 2019). Alternatively, it may be epithelial miR-34a that regulates septal thickening, since miR-34a regulates lung epithelial cell (notably, type II pneumocyte) apoptosis (Syed *et al*, 2017) in experimental BPD. These data validate a role for miR-34a in PDGFRα$^+$ cells in mediating the inhibitory effects of hyperoxia on alveolarization.

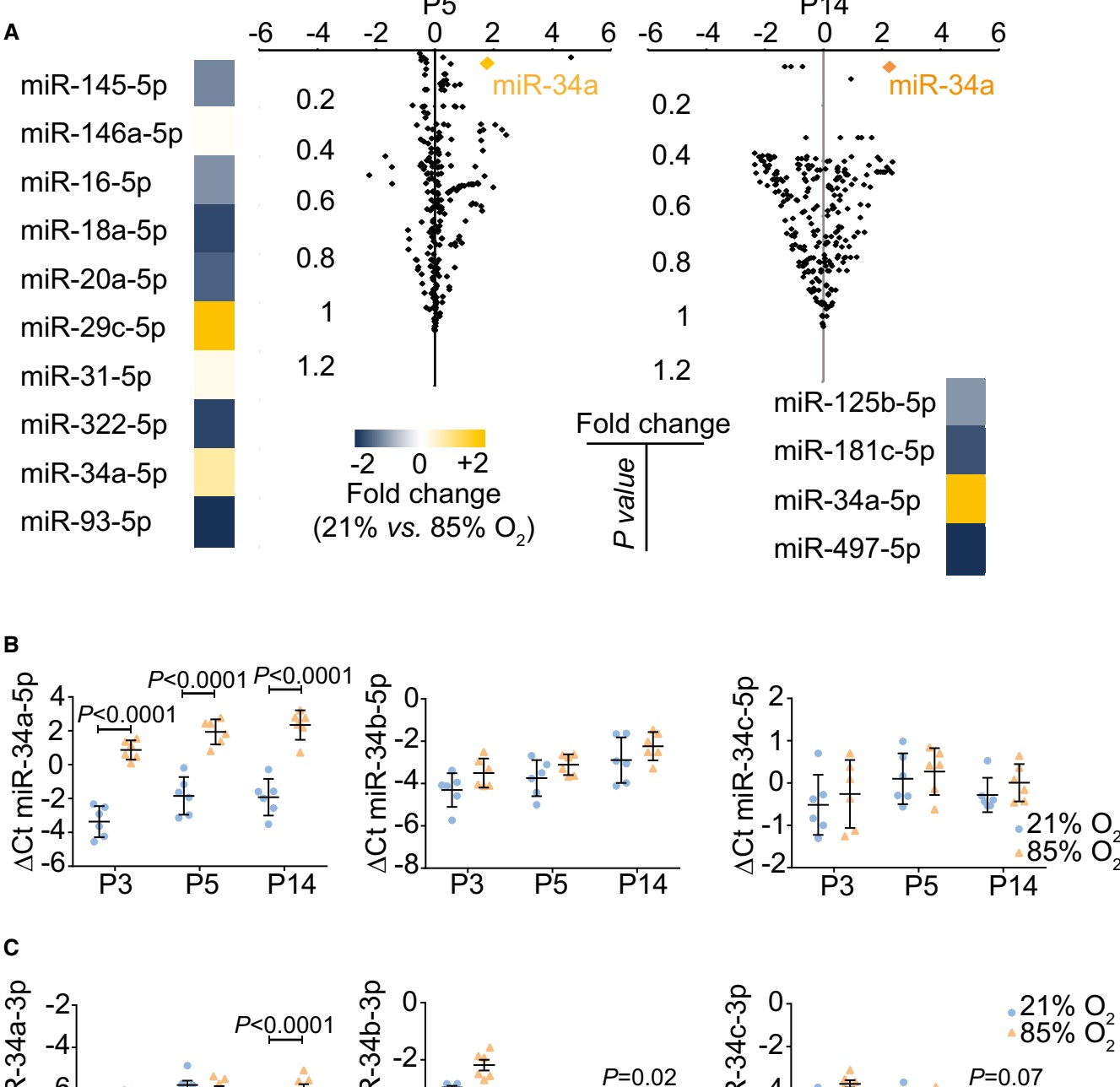

**Figure 1. miR-34a is the most impacted microRNA species in developing mouse lungs after hyperoxia exposure.**

A  Microarray analysis of microRNA expression changes in newborn mouse lungs exposed to 21% $O_2$ versus 85% $O_2$, at post-natal day (P)5 and P14. Microarray data are available at the GEO database under accession number GSE89666.

B  Quantitative RT–PCR detection of microRNA-34a/b/c-5p family members in the lung over the course of normal (21% $O_2$) and aberrant (85% $O_2$) alveolarization.

C  Quantitative RT–PCR detection of microRNA-34a/b/c-3p family members in the lung over the course of normal (21% $O_2$) and aberrant (85% $O_2$) alveolarization.

Data information: For (A), a Welch's approximate *t*-test was employed to determine *P* values (*n* = 4 animals for each experimental group), which were corrected using the algorithm of Benjamini and Hochberg, as described in the Materials and Methods under the heading "Power and statistical analyses". For (B) and (C), data represent mean ± SD (*n* = 6 animals for each experimental group). *P* values were determined by one-way ANOVA with Tukey's *post hoc* modification, and all *P* values < 0.05 for 21% $O_2$ versus 85% $O_2$ comparisons at each developmental stage (P3, P15, and P14) are indicated.

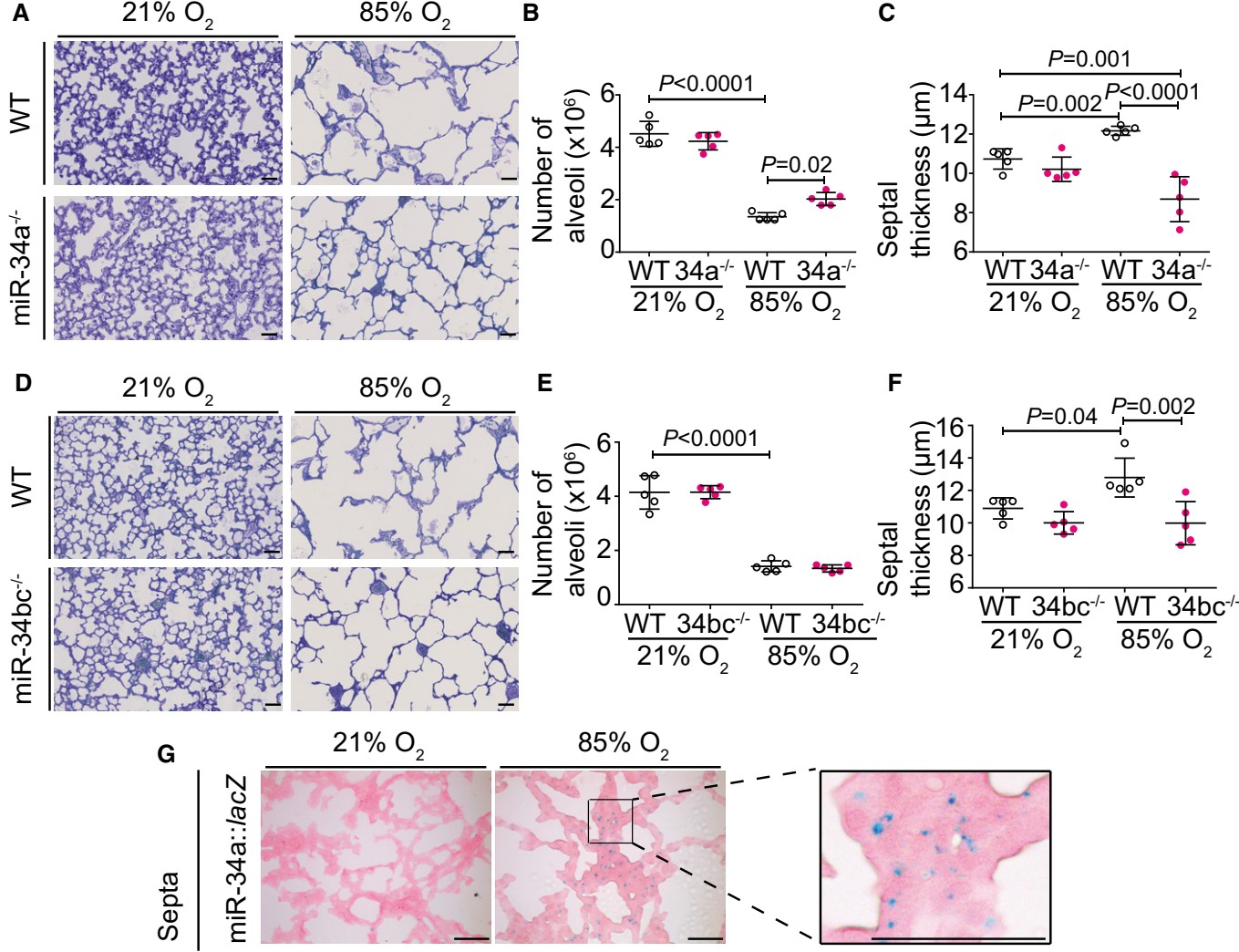

**Figure 2.  miR-34a-5p functionally contributes to arrested lung alveolarization in response to hyperoxia.**

A  Qualitative analysis of lung structure in Richardson-stained plastic-embedded lung sections from wild-type (WT) and miR-34a$^{-/-}$ mice during normal and aberrant alveolarization (scale bar, 50 μm).

B  Quantification of total number of alveoli by design-based stereology in wild-type (WT) and miR-34a$^{-/-}$ mice (34a$^{-/-}$) during normal and aberrant alveolarization.

C  Quantification of mean septal thickness by design-based stereology in wild-type (WT) and miR-34a$^{-/-}$ mice (34a$^{-/-}$) during normal and aberrant alveolarization.

D  Qualitative analysis of lung structure in Richardson-stained plastic-embedded lung sections from wild-type (WT) and miR-34bc$^{-/-}$ mice during normal and aberrant alveolarization (scale bar, 50 μm).

E  Quantification of total number of alveoli by design-based stereology in wild-type (WT) and miR-34bc$^{-/-}$ mice (34bc$^{-/-}$) during normal and aberrant alveolarization.

F  Quantification of mean septal thickness by design-based stereology in wild-type (WT) and miR-34bc$^{-/-}$ mice (34bc$^{-/-}$) during normal and aberrant alveolarization.

G  Localization of miR-34a expression by β-galactosidase activity staining in the developing lungs of P14 miR-34a::*lacZ*$^{+/+}$ mice that were undergoing normal or aberrant alveolarization (scale bar, 50 μm; larger and some additional images are presented in Appendix Fig S2).

Data information: Qualitative data (A, D, G) illustrated from one experiment are representative of the trend observed in four other (A, D) or two other (G) experiments. For all quantitative data sets (B, C, E, F), five animals for each experimental group are illustrated, with each data point representing an individual animal, where data represent mean ± SD. *P* values for selected comparisons were determined by one-way ANOVA with Tukey's *post hoc* modification.

## The miR-34a/*Pdgfra* interaction plays a causal role in aberrant lung alveolarization

MicroRNA/mRNA interactions can be interrupted using target site blocker (TSB) technology. We employed two synthetic TSBs (TSB1 and TSB2) to protect both of the miR-34a-binding sites in the *Pdgfra* 3′-UTR (Fig 4A). Both TSBs protected PDGFRα expression from miR-34a regulation in MLg cells *in vitro* (upper panels, Fig 4B and

C). Both TSBs exhibited specificity for the miR-34a/*Pdgfra* interaction, since neither TSB interfered with the impact of a synthetic miR-34a mimic on levels of c-Kit (middle panel, Fig 4B), a validated miR-34a target (Siemens *et al*, 2013), or of SIRT1 (middle panel, Fig 4C), another validated miR-34a target (Yamakuchi *et al*, 2008). A TSB cocktail of an equimolar TSB1:TSB2 mixture effectively protected PDGFRα expression from miR-34a regulation in MLg cells (Appendix Fig S6). *In vivo*, TSBs afforded some protection against

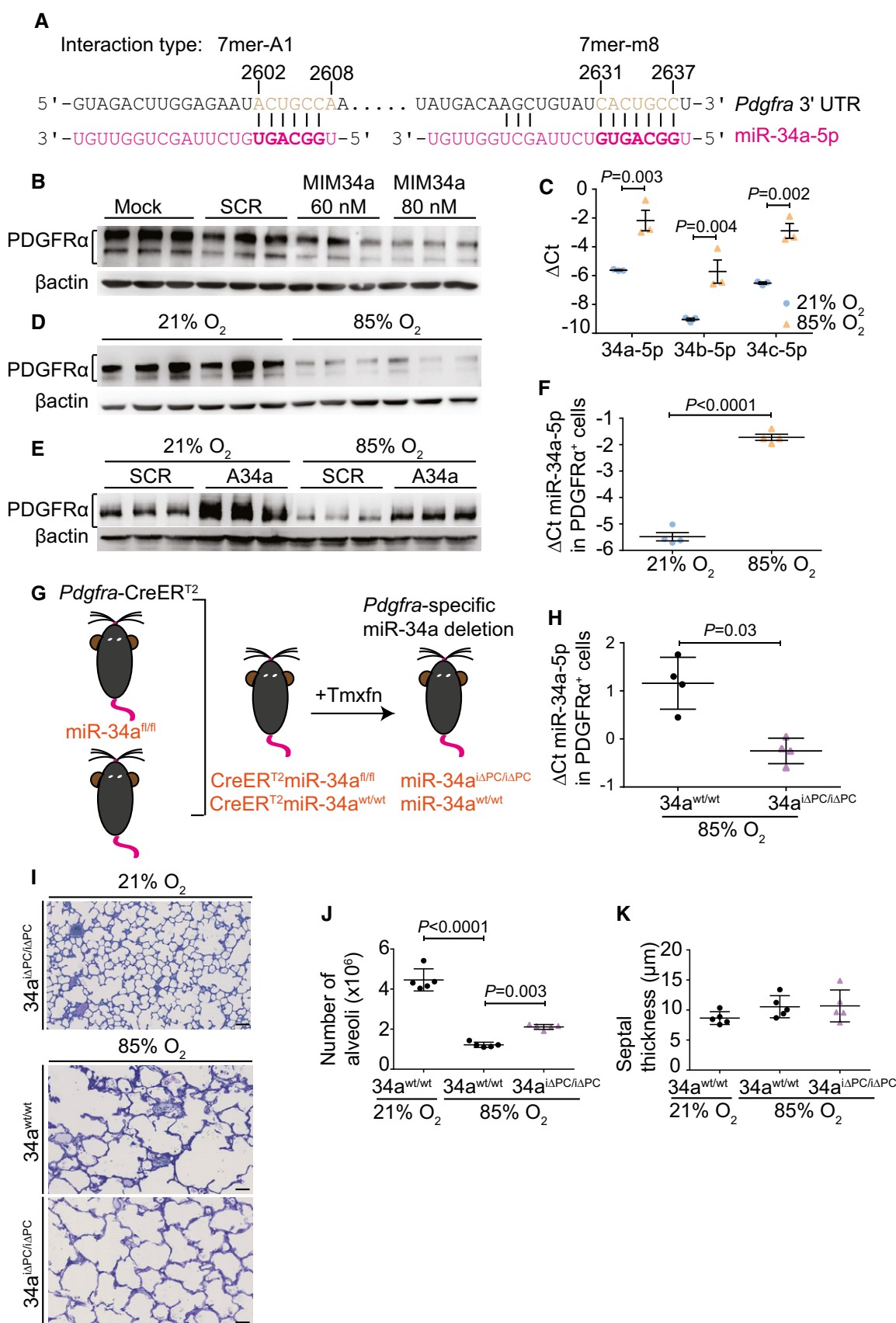

**Figure 3.**

**Figure 3.  miR-34a-5p acts in PDGFRα⁺ cells to block lung alveolarization.**

A   *In silico* identification of miR-34a binding sites in the *Pdgfra* 3′-UTR.

B   Immunoblot detection of PDGFRα levels in MLg cells after treatment with scrambled microRNA (SCR) or a miR-34a (MIM34a) mimic (*n* = 3 separate cell cultures for each group).

C   Quantitative RT–PCR detection of miR-34a/b/c-5p levels in MLg cells *in vitro*, maintained under 21% $O_2$ or 85% $O_2$ (*n* = 3 separate cell cultures for each group).

D   Immunoblot detection of PDGFRα levels in the lungs of mouse pups (*n* = 6 animals for each group) at post-natal day (P)5, during normal (21% $O_2$) and aberrant (85% $O_2$) alveolarization.

E   Immunoblot detection of PDGFRα levels in MLg cells *in vitro*, maintained under 21% $O_2$ or 85% $O_2$, where cells had been transfected wither with a scrambled (SCR) antimiR, or an antimiR directed against miR-34a (A34a) (*n* = 3 separate cell cultures for each group).

F   Quantitative RT–PCR detection of miR-34a-5p levels in PDGFRα⁺ cells, sorted by FACS from the lungs of mouse pups (*n* = 4 animals for each group; data from an independent repetition are provided in Appendix Fig S5) at P5, maintained under 21% $O_2$ or 85% $O_2$ from birth.

G   Schematic illustration of the generation of a conditional, inducible deletion-ready mouse strain, where administration of tamoxifen (Tmxfn) abrogated miR-34a expression in *Pdgfra*-expressing cells (denoted miR-34a^iΔPC/iΔPC^).

H   Quantitative RT–PCR detection of miR-34a-5p levels in PDGFRα⁺ cells, sorted by FACS from the lungs of either wild-type (34a^wt/wt^) mouse pups, or mouse pups in which miR-34a expression in *Pdgfra*-expressing cells (34a^iΔPC/iΔPC^) at P5 (*n* = 4 animals for each group).

I   Qualitative analysis of lung structure in Richardson-stained plastic-embedded lung sections from 34a^wt/wt^ or 34a^iΔPC/iΔPC^ mouse pups at P14 during aberrant (85% $O_2$) alveolarization, compared with 34a^iΔPC/iΔPC^ during normal (21% $O_2$) alveolarization (scale bar, 50 μm). Data are representative of observations made in four other experiments.

J   Quantification of total number of alveoli by design-based stereology in 34a^wt/wt^ or 34a^iΔPC/iΔPC^ mouse pups at P14, during normal and aberrant alveolarization (*n* = 5 animals for each group).

K   Quantification of mean septal thickness by design-based stereology in 34a^wt/wt^ or 34a^iΔPC/iΔPC^ mouse pups at P14, during normal and aberrant alveolarization (*n* = 5 animals for each group).

Data information: For immunoblots (B, D, E), protein loading equivalence was controlled by βactin levels. (C, F, H, J, K) Data represent mean ± SD. In (C, F, and H), *P* values for pair-wise comparisons were calculated by unpaired Student's *t*-test. In (J and K), *P* values for selected comparisons were calculated by one-way ANOVA with Tukey's *post hoc* modification.

Source data are available online for this figure.

the impact of hyperoxia on lung alveolarization in experimental BPD (Fig 4D; Appendix Table S4), where increased alveoli number (Fig 4E) and decreased mean septal thickness (Fig 4F) were noted. Application of the TSB1,2 cocktail increased the abundance of both PDGFRα⁺ cells (Fig 4G; Appendix Fig S7) and PDGFRα⁺/αSMA⁺ myofibroblasts (Fig 4H) in hyperoxia-exposed mouse lungs. These data validate a causal role for the miR-34a/*Pdgfra* interaction in arrested lung development provoked by hyperoxia, most likely through partial restoration of PDGFRα⁺/SMA⁺ myofibroblasts.

MicroRNA function may be modulated *in vivo* using locked nucleic acid (LNA) antimiRs (Patrick *et al*, 2010). We theorized that

dampening functional miR-34a levels in experimental BPD would improve alveolarization; therefore, an antimiR directed against miR-34a was applied therapeutically (concomitantly with hyperoxia exposure; Fig 5A), which decreased functional miR-34a levels in mouse lungs by P5 (Appendix Fig S8A), with no impact on miR-34b, and a moderate impact on miR-34c (Appendix Fig S8B and C). The effect of antimiR-34a on miR-34a was maintained up to P14 (Fig 5B). AntimiR-34a protected alveolarization from hyperoxia (Fig 5C; Appendix Table S5), increasing alveoli number by 40% (Fig 5D), and normalizing septal thickness (Fig 5E). Flow cytometric quantification of PDGFRα⁺ cells and PDGFRα⁺/αSMA⁺ myofibroblasts (analysis

**Figure 4.  Disrupting the miR-34a/*Pdgfra* interaction restores myofibroblast abundance and limits hyperoxic damage to the developing alveolar architecture in mouse lungs.**

A   Generation of two target site blocker (TSB) locked nucleic acid sequences: TSB1 and TSB2 (in blue), for the disruption of the miR-34a/*Pdgfra* interaction, indicating binding sites in the *Pdgfra* 3′-UTR, *Kit* 3′-UTR, and the *Sirt1* 3′-UTR (in black), alongside the miR-34a sequence (in red). The miR-34a seed sequence, and the seed-sequence binding site in the target mRNA 3′-UTR are indicated in bold, and brown, respectively.

B   Evaluation of the specificity of TSB1 and TSB2 in MLg cells using scrambled miR (SCR) and miR-34a (MIM34a) mimics, and probing for PDGFRα and c-Kit as TSB-dependent and TSB-independent target readouts, respectively. Protein loading equivalence was controlled by βactin levels. Note: PDGFRα, βactin, and c-Kit were all probed on the same membrane; hence, a single βactin immunoblot is presented. Data are representative of three experiments.

C   Evaluation of the specificity of TSB1 and TSB2 in MLg cells using scrambled miR (SCR) and miR-34a (MIM34a) mimics, and probing for PDGFRα and SIRT1 as TSB-dependent and TSB-independent target readouts, respectively. Protein loading equivalence was controlled by βactin levels. Note: PDGFRα, βactin, and SIRT1 were all probed on the same membrane; hence, a single βactin immunoblot is presented. Data are representative of three experiments.

D   Qualitative analysis of lung structure in Richardson-stained plastic-embedded lung sections from wild-type mouse pups at post-natal day (P)14, treated with either scrambled target site blocker (SCR) or a cocktail of both target site blockers (TSB1 and TSB2) during normal (21% $O_2$) and aberrant (85% $O_2$) alveolarization (scale bar, 50 μm). Data are representative of three or more experiments.

E   Quantification of total number of alveoli by design-based stereology in wild-type mouse pups at P14, treated with scrambled target site blocker (SCR) or the TSB1,2 cocktail during aberrant alveolarization (*n* = 5 animals for each group).

F   Quantification of mean septal thickness by design-based stereology in wild-type mouse pups at P14, treated with scrambled target site blocker (SCR) or the TSB1,2 cocktail during aberrant alveolarization (*n* = 5 animals for each group).

G   Quantitative analysis of PDGFRα⁺ cells by flow cytometry, in lungs from wild-type mouse pups at P5, treated with scrambled target site blocker (SCR) or the TSB1,2 cocktail during aberrant alveolarization (*n* = 5 animals for each group).

H   Quantitative analysis of PDGFRα⁺/αSMA⁺ cells by flow cytometry, in lungs from wild-type mouse pups at P5, treated with scrambled target site blocker (SCR) or the TSB1,2 cocktail during aberrant alveolarization (*n* = 5 animals for each group).

Data information: Data represent mean ± SD. *P* values were calculated by unpaired Student's *t*-test.

Source data are available online for this figure.

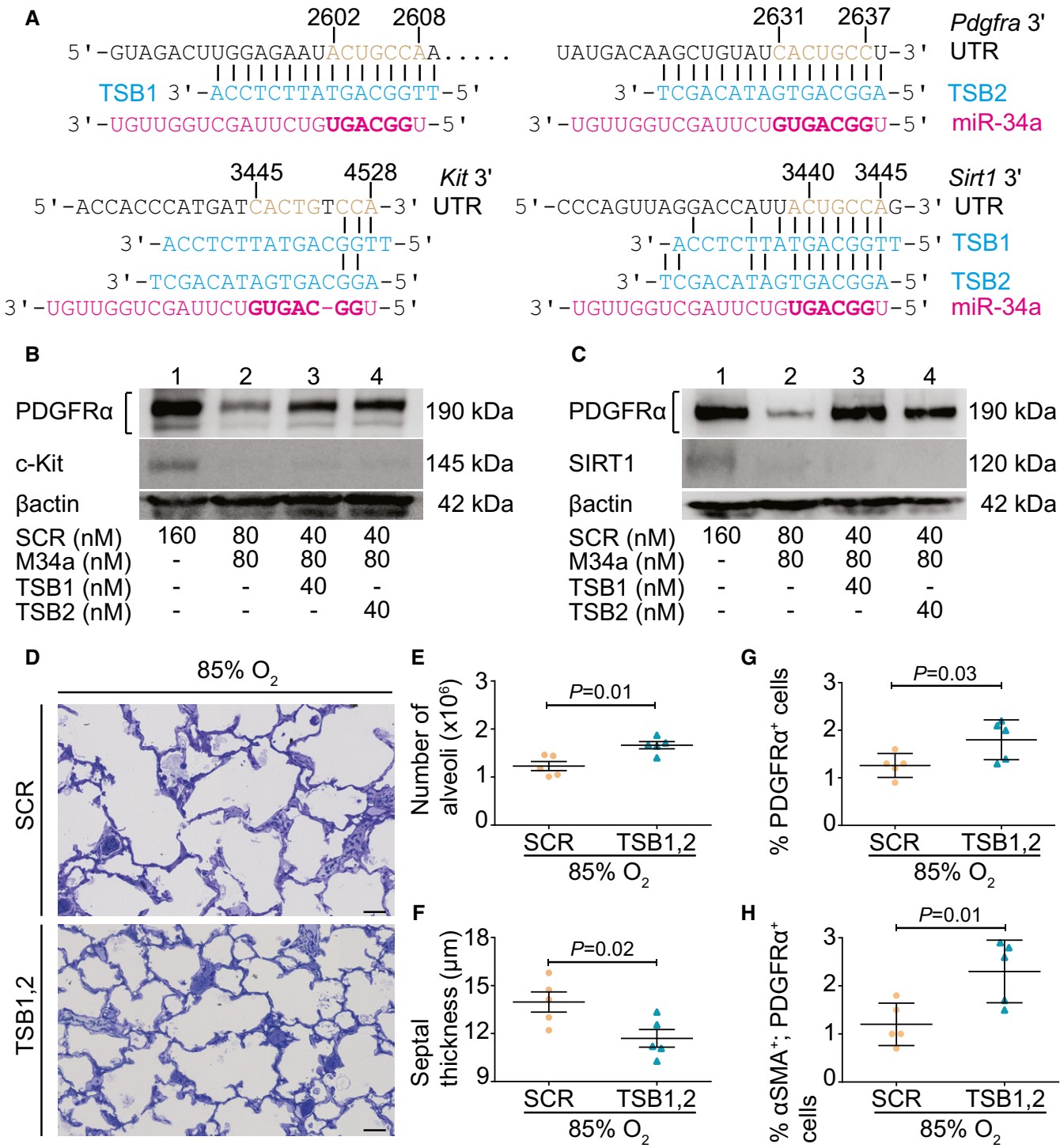

**Figure 4.**

described in Appendix Fig S9A,B), revealed that the abundance of both PDGFRα$^+$ cells (Fig 5F; Appendix Fig S10A) and PDGFRα$^+$/αSMA$^+$ myofibroblasts (Fig 5G; Appendix Fig S10B), both of which were depleted by hyperoxia, was partially restored by antimiR-34a. However, the abundance of αSMA$^+$ cells *per se* was not changed (Appendix Fig S9C and D). These data imply that antimiR-34a partially restored myofibroblast numbers in injured, developing lungs

(schematically presented in Fig 5H). Consistent with this idea, increased elastin foci and improved elastin fiber organization were noted in antimiR-34a-treated mice (Appendix Fig S11).

To further explore the role of hyperoxia and miR-34a on PDGFRα$^+$ cell abundance, apoptosis was assessed in PDGFRα$^+$ cells from hyperoxia-treated mouse pups by flow cytometry (Appendix Fig S12A), where increased apoptosis of PDGFRα$^+$ cells was noted at P5

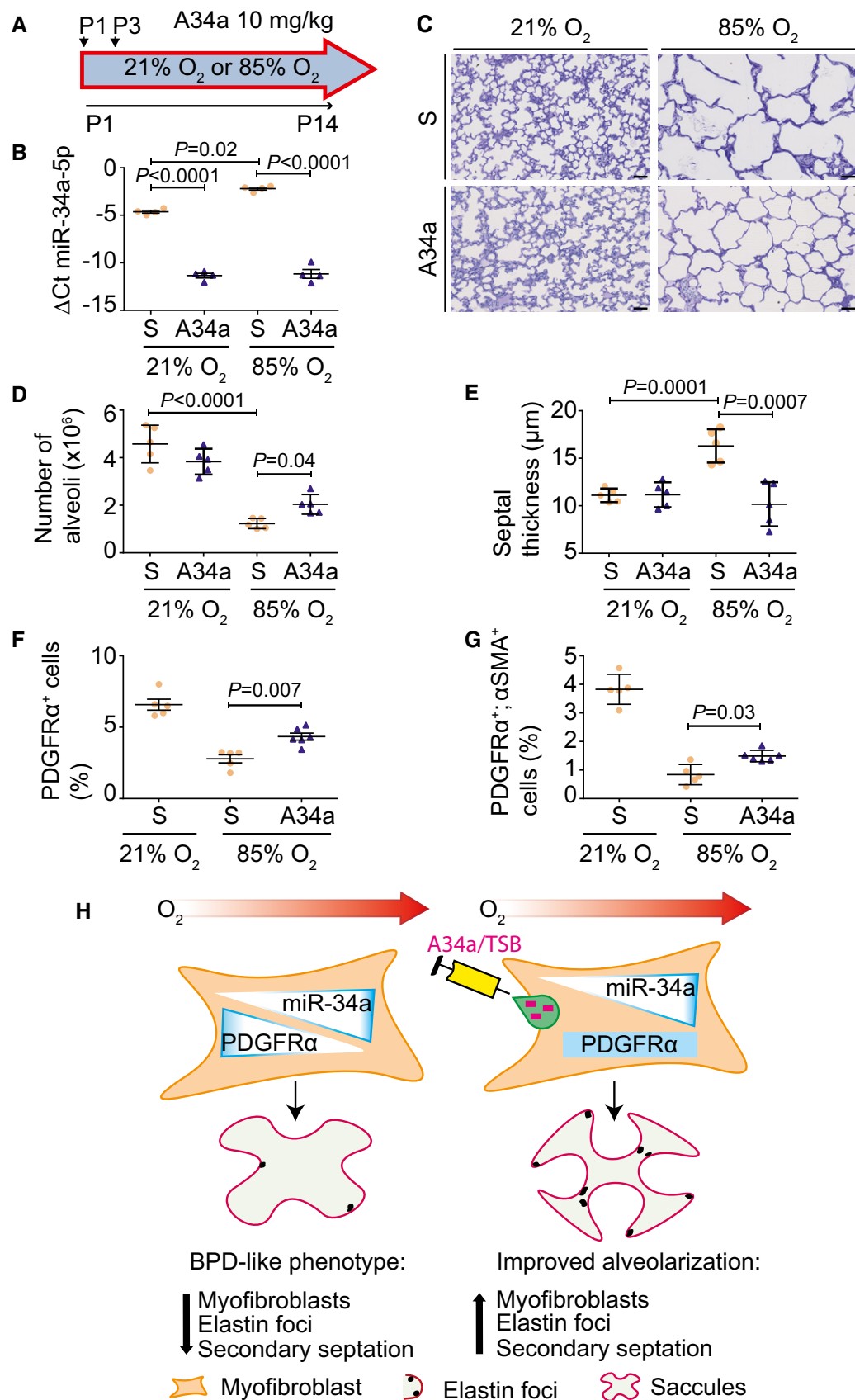

Figure 5.

**Figure 5.  Antagonizing miR-34a function partially restores proper lung alveolarization.**

A  Schematic illustration of the antimiR administration protocol.
B  Quantitative RT–PCR detection of miR-34a-5p levels in wild-type mouse pups at post-natal day (P)14 that had been treated either with a scrambled antimiR (S), or antimiR-34a (A34a), during normal (21% $O_2$) and aberrant (85% $O_2$) alveolarization ($n$ = 4 animals for each group).
C  Qualitative analysis of lung structure in Richardson-stained plastic-embedded lung sections from wild-type mouse pups at post-natal day (P)14, treated with either scrambled antimiR (S), or antimiR-34a (A34a), during normal and aberrant alveolarization (scale bar, 50 μm). Data are representative of three or more experiments.
D  Quantification of total number of alveoli by design-based stereology in wild-type mouse pups at post-natal day (P)14, treated with either scrambled antimiR (S), or antimiR-34a (A34a), during normal and aberrant lung alveolarization ($n$ = 5 animals for each group).
E  Quantification of mean septal thickness by design-based stereology in wild-type mouse pups at post-natal day (P)14, treated with either scrambled antimiR (S), or antimiR-34a (A34a), during normal and aberrant alveolarization ($n$ = 5 animals for each group).
F  Quantitative analysis of PDGFRα⁺ cells by flow cytometry, in lungs from wild-type mouse pups at P5, treated with either scrambled antimiR (S), or antimiR-34a (A34a), during normal and aberrant alveolarization ($n$ = 5 animals for each group).
G  Quantitative analysis of PDGFRα⁺/αSMA⁺ cells by flow cytometry, in lungs from wild-type mouse pups at P5, treated with either scrambled antimiR (S), or antimiR-34a (A34a), during normal and aberrant alveolarization ($n$ = 5 animals for each group).
H  Schematic illustration of the role and translational scope of the miR-34a/*Pdgfra* interaction during arrested lung alveolarization. Hyperoxia drives miR-34a expression in myofibroblasts, downregulating PDGFRα expression and reducing PDGFRα⁺ cell abundance, causing the perturbed elastin fiber production and blunted alveolarization seen in bronchopulmonary dysplasia (BPD). The effects of hyperoxia are attenuated when miR-34a function is blocked with an antimiR (A34a) or when the miR-34a/*Pdgfra* interaction is disturbed with a target site blocker (TSB).

Data information: Data represent mean ± SD. *P* values were calculated by one-way ANOVA with Tukey's *post hoc* modification.

(Appendix Fig S12B), but not at P14 (Appendix Fig S12C). Furthermore, the density of PDGFRα on PDGFRα⁺ cells was reduced by hyperoxia (Appendix Fig S12D), suggesting abundance of PDGFRα in PDGFRα⁺ cells from mouse lungs exposed to hyperoxia (proposed here to be attributable to increased miR-34a levels within those cells). Flow cytometry did not permit an S-phase analysis of cell proliferation due to too few cells per run, as evident in the histogram in Appendix Fig S13, which has insufficiently developed G0/G1 and G2/M peaks. However, immunofluorescence staining of lung cryosections of P5 mice (Fig 6) revealed fewer proliferating PDGFRα⁺ cells in the lungs of hyperoxia-exposed mice at P5 *versus* normoxia-exposed mice. Together, these data indicate that hyperoxia does reduce PDGFRα⁺ cell abundance and proliferation *in vivo* in mice. *In vitro*, primary mouse lung fibroblasts exhibited reduced PDGFRα levels after hyperoxia exposure (Appendix Fig S14A), and increased miR-34a levels after application of a miR-34a mimic (Appendix Fig S14B), consistent with what was noted in MLg cells *in vitro* (Fig 3E). A miR-34a mimic had a moderate impact on baseline proliferation (Appendix Fig S14C) and no impact on baseline apoptosis (Appendix Fig S14D) in primary mouse lung fibroblasts *in vitro*.

Collectively, these data indicate that hyperoxia can decrease proliferation (Fig 6B) and increase apoptosis (Appendix Fig S12B and C) of PDGFRα⁺ cells *in vivo* in developing mouse lungs. Additionally, hyperoxia drives increased levels of miR-34a in PDGFRα⁺ cells that are resident in the developing mouse lung (Fig 3F), which in turn decreases the abundance of PDGFRα in affected cells (Appendix Fig S12D; ostensibly, PDGFRα⁺ myofibroblasts in the developing septa). We propose that this results in defective elastin production and remodeling (one of the functions of myofibroblasts during alveolarization), which in turn impairs secondary septation, leading to alveolar simplification characteristic of BPD (Fig 5H). By neutralizing miR-34a (Fig 5) or disrupting the miR-34a/*Pdgfra* interaction (Fig 4), the abundance of PDGFRα⁺ myofibroblasts was partially restored, leading to partial correction of this alveolarization defect. This is noteworthy given the recent first-in-man report using an antimiR to manage hepatitis C infection, by targeting miR-122 (Janssen *et al*, 2013). We further propose that interventions to block miR-34a function or the miR-34a/*Pdgfra* interaction are candidates for translational development.

**A role for miR-34a in septal thickening as well?**

BPD is also characterized by septal thickening (Jobe, 2016). As a secondary observation in this study, we demonstrate here that miR-34a/b/c impacted septal thinning during alveolarization. Thickened septa arose in this model from multicellular stacking of cells, which revert to the normally observed single cell layer after antimiR-34a treatment (Fig 7). Almost all septal cells stained for aquaporin 5 (Aqp5), a type I pneumocyte marker—a cell type that exhibits tremendous plasticity during alveologenesis (Yang *et al*, 2016)—in both thickened and restored (thinner) septa (Fig 7). In the background of hyperoxia, antimiR-34a treatment did not impact the number or apoptosis (Appendix Fig S15A–G) of type I pneumocytes; or whole-lung gene expression assessed by mRNA microarray at P5 and P14 (Appendix Table S6; GEO accession number GSE89730; validated in Appendix Fig S16). Thus, antimiR-34a most likely affected gene expression in a rare cell population, such as PDGFRα⁺ myofibroblasts, and not broadly throughout the alveolar epithelium, composed largely of type I pneumocytes. We suggest that changes in septal complexity arose not from loss or gain of epithelial cells, but rather from the spatial organization of the type I pneumocytes, that is directed by PDGFRα⁺ myofibroblasts. This may be related to the production of extracellular matrix (ECM) by PDGFRα⁺ myofibroblasts, where perhaps ECM laid down and remodeled during alveologenesis provides migration cues to epithelial cells organize themselves within the newly-generated septa. Such cues might possibly include receptor-mediated interactions between the epithelial cells and the ECM, or matrikine gradients, the latter having been recently implicated in epithelial remodeling in asthma (Patel *et al*, 2018). This general idea is consistent with the observation that PDGFRα⁺ lung fibroblasts decline in number during septal thinning (McGowan & McCoy, 2011) and is in-line with current thinking that epithelial–mesenchymal interactions drive lung development (Hogan *et al*, 2014).

To date, pivotal roles for microRNA processing by Dicer (Harris *et al*, 2006) and Argonaute (Lü *et al*, 2005) in lung branching suggested microRNA control of early (embryonic) lung development (Metzger *et al*, 2008), where functional roles for the miR-17 family have been demonstrated (Carraro *et al*, 2014). In contrast, in late

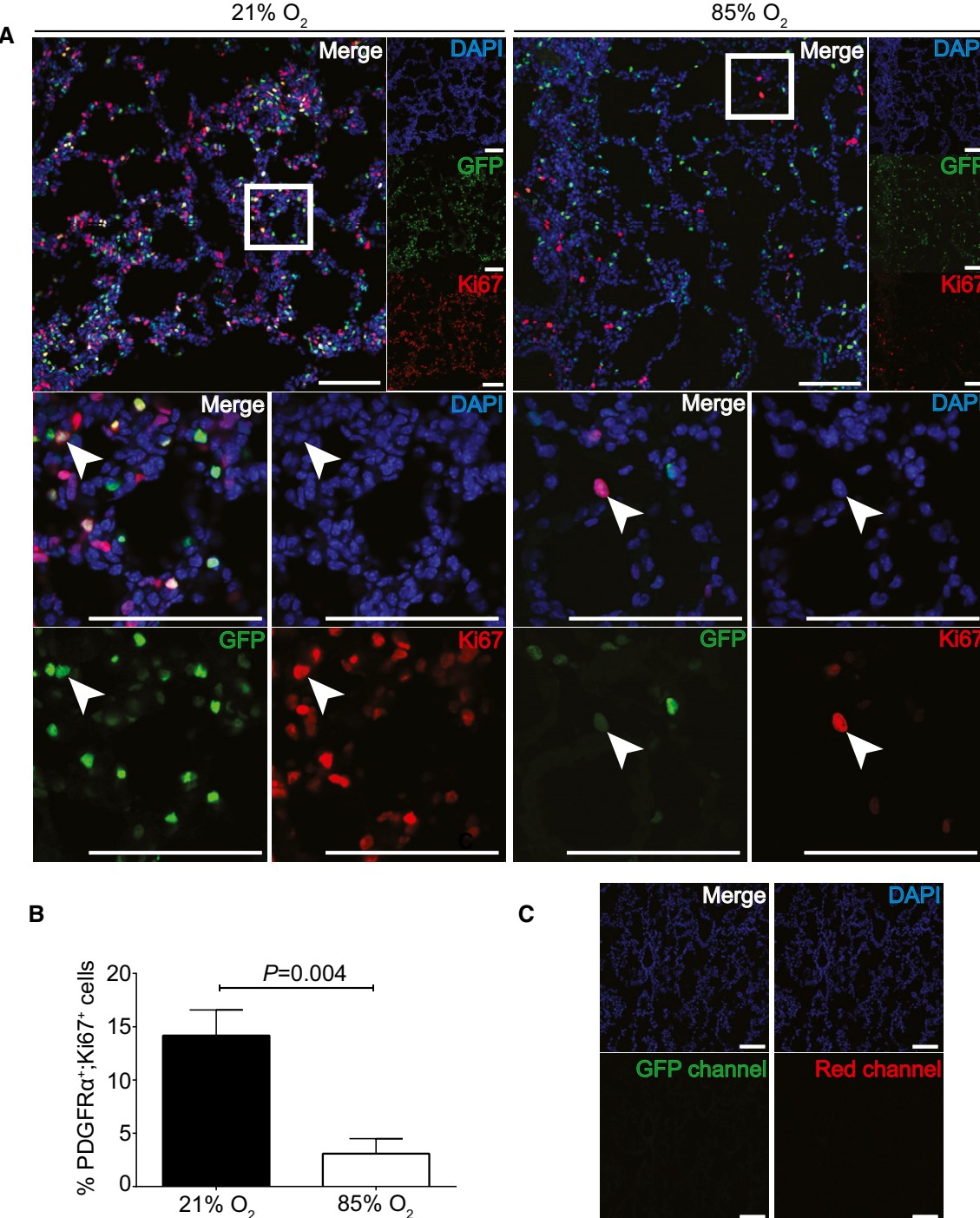

**Figure 6.  Assessment of proliferation status of PDGFRα⁺ cells in developing mouse lungs.**

A    Mice expressing nuclear-localized GFP under the control of the *Pdgfra* promoter were maintained under normoxic (21% O₂) or hyperoxic (85% O₂) conditions, and lungs were harvested, processed, and immunostained for Ki67 to determine proliferation status. DAPI staining revealed nuclei of all cells present in the section. Low-magnification images from individual channels are presented to the right of the merged (large) image first row of images. The area demarcated by the white box in the merge image of the first row is magnified in the second and third rows to allow for visualization of greater magnification of the demarcated region of the merged image, as well as visualization of a single Ki67⁺, GFP⁺ cell (white arrowhead) in all three channels separately. Scale bar: 100 μm.

B    The number of PDGFRα⁺ cells in four microscopic fields was assessed for co-staining with an anti-Ki67 antibody to reveal proliferating cells. *P* values were calculated by unpaired Student's *t*-test (*n* = 4 fields for each group, trends are representative of those observed in two other experiments). Data represent mean ± SD.

C    The Ki67 staining and GFP fluorescence was controlled for by examining lungs from wild-type mice that were treated with an isotype-matched control IgG used for the Ki67 staining experiments. Sections were examined for GFP fluorescence as well as in the red channel used to detect the Ki67 staining. Scale bar: 100 μm. DAPI, 4′,6-diamidino-2-phenylindole; GFP, green fluorescent protein.

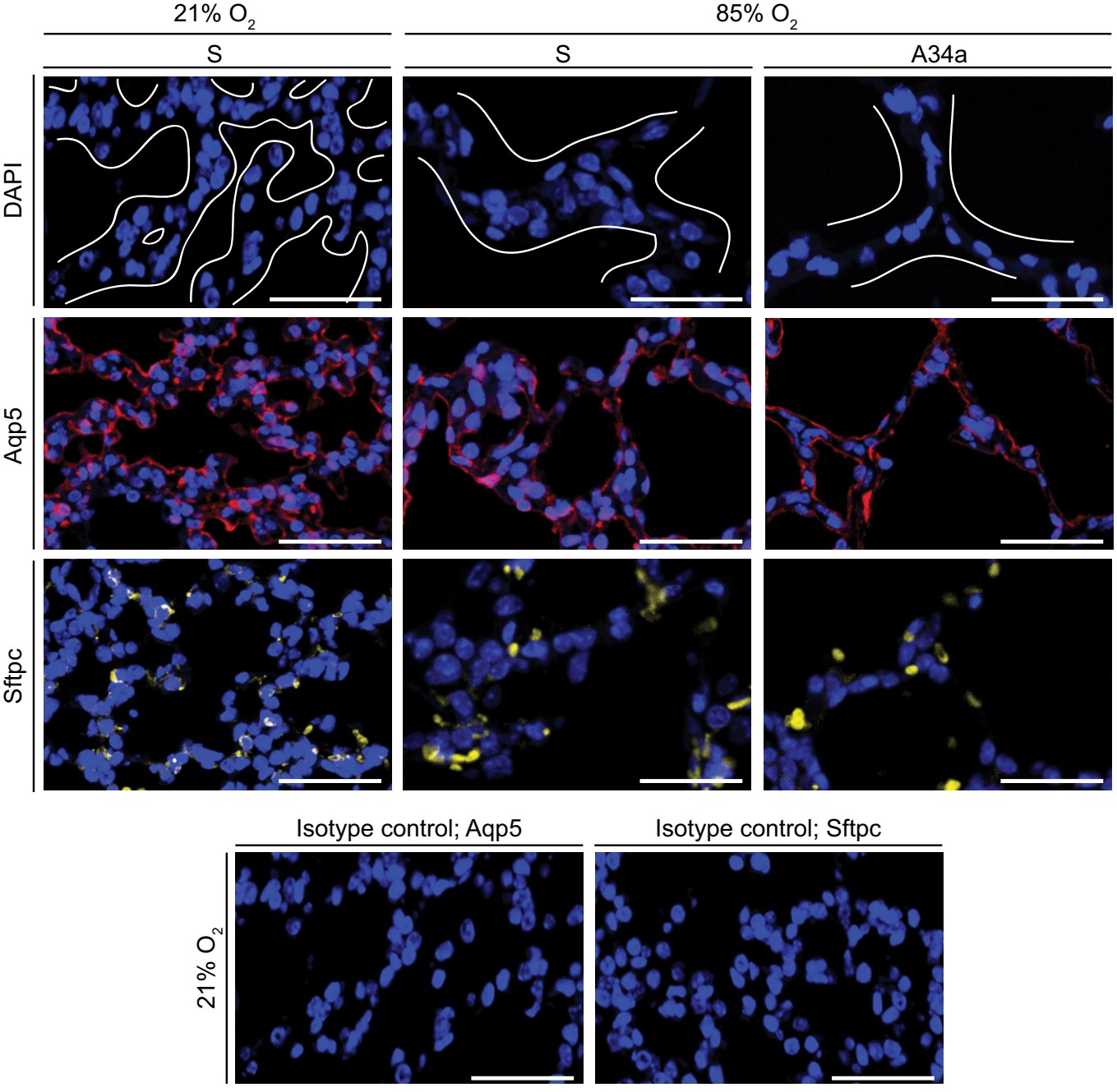

**Figure 7. The primary cell type in the normally and aberrantly developing septa are type I alveolar epithelial cells.**

The impact of administration of scrambled antimiR (S) or an antimiR directed against miR-34a (A34a) on the abundance of type I alveolar epithelial cells (marked by aquaporin 5, Aqp5) and type II alveolar epithelial cells (marked by pro-surfactant protein C, Sftpc) was assessed in 3-µm sections of paraffin-embedded lung tissue from P5 mice undergoing normal (21% $O_2$) or aberrant (85% $O_2$) lung alveolarization. DAPI, 4′,6-diamidino-2-phenylindole. In the DAPI images, white lines delineate tissue from airspaces, and in the 85% $O_2$ groups demarcate septa. Antibody specificity was validated by rabbit IgG isotype control primary antibodies. The control experiments for the Aqp5 and Sftpc staining runs are illustrated here. Scale bars, 50 µm.

lung development, which is relevant to BPD, several microRNA candidates have been proposed as pathogenic players, including miR-150 (Narasaraju *et al*, 2015), miR-489 (Olave *et al*, 2016), miR-29b (Durrani-Kolarik *et al*, 2017), the miR-19/72 cluster (Rogers *et al*, 2015; Robbins *et al*, 2016), and epithelial miR-34a (Syed *et al*, 2017), but transgenic mouse studies have only validated a causal

role for epithelial miR-34a (most likely by targeting angiopoietin) in arrested alveolarization, where miR-34a levels were also documented to be elevated in the lungs of BPD patients (Syed *et al*, 2017). Ours is the first report of a causal role being validated for any microRNA/mRNA target interaction in aberrant lung alveolarization, as well as the first-in-mouse use of a TSB *in vivo* in an animal

model of human disease. Notably, translation of these findings into the use of antimiR treatment of mice in the BPD model documented marked benefit in this preclinical model. We, therefore, highlight a potentially druggable pathway to manage arrested alveolarization following preterm birth.

# Materials and Methods

### Regulatory authority compliance and legal approvals

Animal experiments reported in this study were approved by the *Regierungspräsidium Darmstadt*, under approval numbers B2/277, B2/1002, and B2/1060.

### Mice

Wild-type *Mus musculus* C57Bl/6J mice were obtained from The Jackson Laboratory. The generation and characterization of a tamoxifen-inducible *Pdgfra*-Cre driver mouse strain [Tg(Pdgfra-cre/ERT2)1Wdr; MGI:3832569] referred to herein as *Pdgfra*-CreER^T2, on a C57BL/6J background has been described previously (Rivers *et al*, 2008; Ntokou *et al*, 2015). A *lacZ*-tagged miR-34a gene-trap strain (Mir34a^tm1.1Lhe; MGI:5308792) (Choi *et al*, 2011), herein referred to as miR-34a::*lacZ* or miR-34a^−/− (on a C57BL/6J background) was always employed in the homozygous state and was used interchangeably as a miR-34a global knockout and a miR-34 *lacZ* reporter, was obtained from the Jackson Laboratory. A miR-34bc global knockout strain (Mir-c21^tm1.1Lhe; MGI:5308794) (Concepcion *et al*, 2012) on a C57BL/6J background was always employed in a homozygous state, is referred to herein as miR-34bc^−/−, and was obtained from the Jackson Laboratory. A mouse strain expressing a human histone 2B-enhanced green fluorescent protein fusion protein under the control of the *Pdgfra* promoter (B6.129S4-*Pdgfra*^tm11(EGFP)Sor/J; MGI:3766768) (Hamilton *et al*, 2003) was obtained from the Jackson Laboratory, and was always employed in the heterozygous state, and allowed the detection of *Pdgfra*-expressing cells through nuclear-localized GFP fluorescence. A strain carrying a floxed miR-34a allele (Mir34a^tm1.2Aven; MGI: 5320795) (Concepcion *et al*, 2012) on a C57BL/6J background was always employed in the homozygous state, is referred to herein as miR-34a^fl/fl, and was obtained from the Jackson laboratory. A conditional, inducible deletion-ready strain, where administration of tamoxifen can abrogate miR-34a expression exclusively in *Pdgfra*-expressing cells (denoted miR-34a^iΔPC/iΔPC), was created by crossing the *Pdgfra*-CreER^T2 driver strain with a miR-34a^fl/fl strain. This strain was always employed on a C57BL/6J background with the *Pdgfra*-CreER^T2 allele in the heterozygous state, and the floxed miR-34a allele in the homozygous state. Studies employing the miR-34a^iΔPC/iΔPC strain were controlled for with a strain heterozygous for *Pdgfra*-CreER^T2 but carrying two wild-type miR-34a alleles. For induction of tamoxifen-responsive genes, a protocol has been developed and validated that allows for the tamoxifen treatment of newborn pups under hyperoxic conditions, where tamoxifen is poorly tolerated (Ruiz-Camp *et al*, 2017): Newborn pups received a single intraperitoneal injection on the day of birth [post-natal day(P)1)] of 0.2 mg tamoxifen/pup in 10 μl Miglyol 812. All mice were maintained on a 1320 formula maintenance diet for rats and mice (Altromin), available *ad libitum* together with drinking water, with as 12 h:12 h day/night cycle.

### The hyperoxia-based mouse model of bronchopulmonary dysplasia

Bronchopulmonary dysplasia was modeled in mice in a protocol well established in our laboratory (Nardiello *et al*, 2017b) where newborn mouse pups, randomized to litters of equal numbers of pups per nursing dam, are exposed to 85% $O_2$ from P1 to P14, while control mouse pups with normal lung development are exposed in parallel to 21% $O_2$. In the case of tamoxifen-induced gene expression where mice received tamoxifen on P1, hyperoxia exposure was initiated on P2. Both male and female animals were used, since no sex bias has been noted in studies on perturbations to lung development of C57Bl/6J mice in response to hyperoxia (Nardiello *et al*, 2017b). Nursing dams were rotated between normoxia and hyperoxia at 24-h intervals, to limit oxygen toxicity. At either P3 (prior to bulk lung alveolarization), at P5 (the peak period of bulk lung alveolarization), or at P14 (after completing of the bulk alveolarization phase), mice were killed by pentobarbital overdose (500 mg/kg, intraperitoneal) and lungs were removed *en bloc* for further analysis. The investigators were not blinded to group allocation, but were blinded to outcome assessment.

### Microarray analyses

For an unbiased analysis of microRNA expression over the course of normal and aberrant lung development, microRNA was isolated with a miRNeasy Mini kit (Qiagen), and microRNA expression was assessed using an Agilent-035430 mouse miRNA array platform (miRBase release 17 miRNA ID version; Mouse_8x60K-v17). For an unbiased analysis of mRNA expression over the course of normal and aberrant lung development after antimiR administration, mRNA was isolated with a peqGOLD total RNA kit (Peqlab), and mRNA expression was assessed using an Agilent-028005 SurePrint G3 Mouse GE 8 × 60K Microarray platform. Microarray analyses were undertaken by IMGM Laboratories (Munich).

### Gene and protein expression analysis

Changes in gene expression were assessed by SYBR green-based real-time RT–PCR (using *Rnu6* and *Polr2a* as a reference for microRNA and mRNA, respectively) as described previously, after miRNA isolation with a miRNeasy Mini kit (Qiagen) (Hönig *et al*, 2018) or mRNA isolation with a peqGOLD total RNA kit (Peqlab) (Alejandre-Alcázar *et al*, 2007). For microRNA analysis, primer mixtures were purchased from Qiagen: miR-34a-3p (MS00025697), miR-34a-5p (MS00001428), miR-34b-3p (MS00011900), miR-34b-5p (MS00007910), miR-34c-3p (MS00011907), and miR-34c-5p (MS00001442). The primers used for RT–PCR and genotyping PCR analyses are described in Appendix Tables S7 and S8, respectively. The real-time RT–PCR data are presented as the difference in cycle threshold (CT), ΔCT, which reflects the $CT_{(reference\ gene)}–CT_{(gene\ of\ interest)}$. Changes in protein expression were assessed by immunoblot (using βactin or GAPDH to demonstrate loading equivalence), after protein isolation from lung tissue in a Precellys 24-Dual homogenizer (Peqlab) as described previously (Mižíková *et al*, 2015), or protein isolation of cultured cells in Nonidet P-40-containing lysis buffer, as described previously (Madurga *et al*, 2014). The primary antibodies used for immunoblotting are described in Appendix Table S9, and blots

were developed either with a donkey anti-goat HRP-conjugated (Santa Cruz, sc-2020; 1:2,500) or goat anti-rabbit (Thermofisher, 31460; 1:3,000) secondary antibody.

## Stereological analysis of lung structure

Lung structure was assessed by design-based stereology with systemic uniform random sampling, on mouse lungs that were pressure fixed at 20 cm $H_2O$, and treated with arsenic, osmium and uranium, and embedded in plastic (Technovit 7100) resin, sectioned at 2 μm, stained with Richardson's stain, and image captured in a Nanozoomer-XR C12000 (Hamamatsu), exactly as described previously (Madurga *et al*, 2014; Mižíková *et al*, 2015; Nardiello *et al*, 2017b). Lung volume was determined by the Cavalieri principle (Madurga *et al*, 2014). Stereological analyses were undertaken using the NewCast PLUS version VIS4.5.3. computer-assisted stereology system (Visiopharm) and facilitated the determination of *inter alia* total number of alveoli in the lung, the mean septal thickness, and total gas-exchange surface area.

## In situ β-galactosidase activity detection

Cryosections (10 μm) from developing mouse lungs attached to glass microscope slides were fixed in 0.5% (m/v) glutaraldehyde in PBS (10 min, 4°C), washed (by immersion in 1 mM $MgCl_2$ in PBS, 2 × 15 min, RT), and pre-incubated in 5-bromo-4-chloro-3-indolyl-β-D-galactopyranoside (X-Gal) buffer [5 mM potassium ferrocyanide (II), 5 mM potassium ferricyanide (III), 1 mM $MgCl_2$ in PBS, pH 7.0; 10 s, RT] followed by incubation overnight with 1 mg/ml X-Gal in X-Gal buffer at 37°C in the dark. Slides were then washed with 1 mM $MgCl_2$ in PBS (15 min, RT), followed by fixation in 4% (m/v) paraformaldehyde in PBS (4 min); and dehydrated in a graduated ethanol series [100% (v/v), 96% (v/v), 70% (v/v); 5 min each, RT], washed by immersion in PBS (5 min, RT), followed by eosin counterstaining 1% (m/v) in $dH_2O$:ethanol 20:80 (30 s, RT). Slides were washed by immersion in $dH_2O$ (2 s, RT) and mounted with PERTEX (Histolab). Sections were examined using a Leica DM6000B light microscope (Leica).

## Cell isolation and cell culture

Primary mouse lung fibroblasts were isolated from C57Bl/6J mice. Briefly, lungs were instilled with approximately 500 μl of preheated (37°C) collagenase type I (2 mg/ml; Sigma-Aldrich), and subsequently excised *en bloc* from adult female C57BL/6J mice that were killed by isoflurane inhalation. Lungs were placed in 50-ml Falcon tubes containing 25 ml of preheated (37°C) collagenase type I and incubated on a Unimax1010 orbital rotator with gentle agitation at 70 r.p.m, for 1 h at 37°C. Lungs were minced with sterile scissors, and the tissue suspension was dispersed by repeated gentle passage through a 20G syringe needle. The cell suspension was then passed through a 40-μm filter into a new 50-ml Falcon tube. The cell suspension was centrifuged at 120 × *g* for 8 min at 4°C. The supernatant was discarded, and the cell pellet was resuspended in 5 ml of pre-warmed (37°C) high-glucose DMEM containing 10% (v/v) FCS, 100 U/ml penicillin (ThermoFisher), 100 μg/ml streptomycin (ThermoFisher), and seeded into a T-75 cell culture flask (1 flask per lung) and passaged in low-glucose DMEM containing 10%

(v/v) FCS, 100 U/ml penicillin (ThermoFisher), 100 μg/ml streptomycin (ThermoFisher). Primary mouse lung fibroblasts were employed throughout this study, with the exception of *in vitro* hyperoxia exposure and when Lipofectamine® 2000 was used as transfection reagent. In the latter two cases, the MLg mouse lung fibroblast cell line (ATCC® CCL-206™) was employed, and was obtained from the American Type Culture Collection, and maintained in EMEM supplemented with 10% (v/v) FBS. Cultures of primary cells and cell lines were routinely (monthly) screened for mycoplasma contamination.

## MicroRNA mimic, antimiR, and target site blocker interventions *in vitro* and *in vivo*

A synthetic scrambled miR mimic and a miR-34a mimic (catalog numbers SI03650318 and MSY0000542, respectively; Qiagen) were transfected into primary mouse lung fibroblasts with HiPerFect (Qiagen) or MLg cells with Lipofectamine® 2000 or 3000, as per manufacturer's instructions. Locked nucleic acid (LNA) oligonucleotides (purchased from Exiqon) included a scrambled (inert) sequence (5′-ACGTCTATACGCCCA-3′); an antimiR directed against miR-34a (5′-AGCTAAGACACTGCC-3′) and miRCURY LNA™ microRNA Target Site Blockers (herein referred to as target site blockers) directed to target the interaction between the two miR-34a binding sites in the mouse *Pdgfra* 3′-UTR and miR-34a: 5′-TTGGCAGTATTCTCCA-3′ (TSB1) and 5′-AGGCAGTGATACAGCT-3′ (TSB2) (see Fig 3A). *In vitro*, synthetic oligonucleotides were transfected into MLg cells with Lipofectamine® 2000. When combined, synthetic microRNA mimics and LNA target site blockers were applied together as a cocktail, at a final cumulative concentration of 160 nM (Fig 4B). *In vivo*, both target site blockers (applied as a cocktail of a 1:1 mixture of TSB1 and TSB2) and a scrambled or miR-34a-specific antimiR were all applied by intraperitoneal injection at a dose of 10 mg/kg at P1 and P3, in $ddH_2O$.

## Flow cytometry and FACS

All flow cytometry protocols and gating strategies are indicated in the relevant supplementary figures in the Appendix. Antibody conjugates, dilutions, and commercial sources are detailed in Appendix Table S9. Flow cytometry was performed to estimate apoptosis (by annexin V staining) and to quantify cell populations in developing mouse lungs, using the antibodies listed in Appendix Table S9. Flow cytometry and FACS were performed with an LSR Fortressa or an FACSAria III cell sorter, respectively, operated with DIVA software (BD Bioscience). Single-cell suspensions were prepared from mouse pup lungs by instilling approximately 300 μl of Dispase (50 U/ml; BD Bioscience) followed by incubation for 30 min at 37°C. Lungs were dissociated in a gentleMACS™ Dissociator (Miltenyi) in 5 ml (per lung) DMEM supplemented with 10% (v/v) FCS, 100 U/ml penicillin (ThermoFisher), 100 μg/ml streptomycin (ThermoFisher), and 320 U/ml bovine pancreatic DNAse 1 (Serva). To remove cell debris and blood clots, whole-lung cell suspensions were filtered through 100 μm and 40-μm filters. After centrifugation at 266 × *g* for 10 min at 4°C, cell pellets were resuspended in Flow Cytometry Staining Buffer (eBioscience; 00-4222-26), blocked with 1:100 Mouse BD Fc Block™ (BD Biosciences), and incubated with the appropriate primary antibodies or

isotype controls diluted in Flow Cytometry Staining Buffer for 20 min at 4°C in the dark. After washing with Flow Cytometry Staining Buffer, whole-lung cell suspensions were incubated with secondary antibodies diluted in Flow Cytometry Staining Buffer for 20 min at 4°C in the dark. In the case of intracellular staining for myofibroblasts, cells were fixed in 0.15% (m/v) paraformaldehyde in PBS for 10 min at 4°C and permeabilized with 0.2% (m/v) saponin (Calbiochem) diluted in PBS for 15 min at 4°C prior to addition of antibodies. For assessment of epithelial cell apoptosis, stainings were carried out on fresh non-permeabilized cells, where cells were washed and resuspended in annexin V buffer (BD Biosciences) prior to annexin V-Alexa Fluor 647 conjugate (Thermo-Fisher A23204; 1:100) incubation in annexin V buffer for 20 min at 4°C in the dark. Where a fluorophore-conjugated secondary antibody was not employed, cells that had been labeled with an unconjugated primary antibody were treated with a biotin-conjugated secondary antibody, followed by a Streptavidin-phycoery-thrin conjugate (Biolegend 405204; 1:300). For S-phase analysis, live-cell determinations were made by incubation of cell suspensions Hoechst 33342 (Sigma B2261, 5 µg/ml) in PBS for 45 min at 37°C in the dark. For assessment of PDGFRα$^+$ cell apoptosis, live cell single-cell suspensions were prepared as described above, up to and including the step employing Mouse BD Fc Block™, after apoptosis was detected in cell suspensions with an Annexin V kit (Biolegend, 640906).

For isolation of PDGFRα$^+$ cells by cell sorting, the anti-PDGFRα-APC conjugate was coupled to microbeads using 30 µl of anti-APC MicroBeads (Miltenyi 130-097-143) for 20 min at 4°C in the dark, and separated in a AutoMacs separator (Miltenyi) prior to cell sorting. The RNAqueous-Micro kit (Thermo Fisher) was employed to isolate mRNA from PDGFRα$^+$ cells, which are present in low number. To exclude dead cells and debris, 1 µl of 5 mg/ml DAPI or 5 µl of 50 µg/ml 7-ADD (Biolegend) was pipetted into the whole-lung cell suspensions just before the cell analysis.

### Histochemistry and immunofluorescence

Histochemical staining for elastin was undertaken on 3-µm sections from P14 mice, exactly as described previously (Mižíková *et al*, 2015). Immunofluorescence analysis of aquaporin 5 (for type I alveolar epithelial cells), pro-SP-C (for type II alveolar epithelial cells), and 4′,6-diamidino-2-phenylindole (DAPI; to detect cell nuclei) staining was undertaken as described previously (Ntokou *et al*, 2015), using the primary antibodies listed in Appendix Table S9, and Alexa Fluor 647-conjugated goat anti-rabbit IgG (Thermofisher, A21245; 1:500) secondary antibody. Briefly, prior to antibody application, paraffin-embedded sections mounted on glass slides were dehydrated by immersion in Roti®-histol (Roth) (3 × 10 min), followed by a graduated alcohol series [100% (m/v) ethanol for 2 × 5 min; 96% (m/v) ethanol for 1 × 5 min; 70% (m/v) ethanol for 1 × 5 min; PBS for 3 × 5 min], after which antigen retrieval was performed with 10 mM citrate buffer, pH 6, containing 0.05% (v/v) Tween-20, by boiling for 10 min followed by cooling at room temperature over 30 min. Sections were washed (2 × 5 min) in PBS, followed by blocking in 50% (v/v) goat serum in primary antibody buffer [0.5% (v/v) Triton X-100, 0.1% (m/v) bovine serum albumin in PBS]. Primary and secondary antibodies were applied in primary antibody buffer, overnight (at 4°C) or for 1 h (at RT),

respectively. Prior to mounting in Mowiol® 4-88 (Sigma), sections were incubated with 0.005 mg/ml DAPI for 10 min at RT. Images were captured in *Z*-stacks using a LSM-710 confocal microscope (Zeiss).

For the assessment of cell proliferation by immunofluorescence, cryosections were prepared from P5 mouse lungs, where lungs were exposed by midline thoracotomy, perfused transcardially with 1× PBS, and inflated with 1:1 PBS: Tissue-Tek® O.C.T. (Sakura, 4583), removed *en bloc* from the thorax, and frozen at −20°C. Frozen tissue was sectioned at 10 µm with a cryostat; sections were mounted on glass slides and stored at −20°C before fixation. Frozen sections were fixed with cold (−20°C) 4% (m/v) paraformaldehyde (15 min, room temperature) and blocked with normal goat serum diluted 1:1 with 3% (m/v) BSA dissolved in 1× PBS containing 0.3% (v/v) Triton X-100 (1 h at room temperature). Sections were permeabilized with 1% (m/v) saponin in 1× PBS (20 min at room temperature). Ki67 was detected using an anti-Ki67 primary antibody (Appendix Table S9), and an Alexa Fluor 647-conjugated goat anti-rat (Invitrogen, A21247; 1:500) secondary antibody followed by incubation in 4′,6-diamidino-2-phenylindole (DAPI; 1:1,000 dilution of a 1 mg/ml stock solution in PBS). *Z*-stack images of the sections were acquired using a Zeiss LSM710 Laser Scanning Confocal Microscope. For enumeration of Ki67$^+$ cells, the number of Ki67$^+$ cells and GFP$^+$ cells was assessed in a total of 500 DAPI$^+$ cells, per microscopic field; and four fields were assessed per experimental condition.

### Assessment of apoptosis and cell proliferation *in vitro*

Primary mouse lung fibroblasts were seeded at 4,000 cells (in 100 µl) per well of a 96-well tissue culture plate (Greiner, 655180, for proliferation; Greiner, 655098, for apoptosis), incubated overnight, and starved in serum-free OptimMEM (Gibco, 31985-062) for 1 h.

For assessment of proliferation, cells were transfected either with a scrambled microRNA mimic or a miR-34a-5p mimic (80 nM final concentration, as described above), and proliferation was monitored by BrdU incorporation using a colorimetric Cell Proliferation ELISA kit (Roche, 11647229001) after a 1-h serum starvation period in DMEM GlutaMAX (Gibco, 21885-025), followed by 24 h in DMEM GlutaMAX supplemented with 10% (v/v) FCS and 1% Penicillin-Streptomycin solution. Signal was allowed to develop over 5–30 min, as was read in an Infinite M200 Pro spectrophotometer (Tecan).

For assessment of apoptosis, cells were transfected with a scrambled microRNA mimic or a miR-34a-5p mimic as described for proliferation, above, and caspase 3 and caspase 7 activity was detected as a surrogate for apoptosis, using a Caspase-Glo® 3/7 Assay System (Promega, G8091) after 24 h. For a positive control, medium was supplemented with staurosporine (Cayman Chemical, 62996-74-1; 0.5 µM) for the last 6 h of the 24-h period. Luminescence was determined for 60 min, in an Infinite M200 Pro luminometer (Tecan).

### Power and statistical analyses

A prospective power analysis was undertaken for all animal studies, to assess the sample size required. Samples sizes were calculated

using G*Power 3.1.9.2 (Faul *et al*, 2007). For changes in microRNA and mRNA expression assessed by real-time RT–PCR in mouse lung homogenates, a $\Delta Ct$ of |0.5| was considered relevant, resulting in an effect size of $d = 2.70$ (using miR-34a-5p expression as reference values), where $d$ is Cohen's effect size, and using $\alpha = 0.5$ (where $\alpha$ is the Type I error), and a power (1-$\beta$) of 0.8, where $\beta$ is the type II error; required a sample size of $n = 4$ animals per group. For cells sorted by FACS from mouse lungs and processed for microRNA or mRNA analyses, where a pronounced change in gene expression was anticipated, a $\Delta Ct$ of |1.0| was considered relevant, resulting in an effect size of $d = 2.79$ (using miR-34a-5p levels in FACS-sorted PDGFR$\alpha^+$ cells as reference values), and using $\alpha = 0.5$ and a power (1-$\beta$) of 0.8, required a sample size of $n = 4$ animals per group. For changes in rare cell populations in single-cell suspensions from whole lungs of mice, assessed by flow cytometry, a doubling of the cell population (100% increase) was considered relevant, resulting in an effect size of $d = 2.88$ (using PDGFR$\alpha^+$ cell abundance in whole-lung suspensions as reference values), and using $\alpha = 0.5$ and a power (1-$\beta$) of 0.8, required a sample size of $n = 4$ animals per group. Assessment of lung structure included two parameters (total number of alveoli in the lung and mean septal thickness), both assessed by design-based stereology in the same lungs from the same animals. For assessment of total number of alveoli, a 50% increase in the total number of alveoli was considered relevant, resulting in an effect size of $d = 4.13$ (using the hyperoxia-treated wild-type mouse lungs as reference values), and using $\alpha = 0.5$ and a power (1-$\beta$) of 0.8, required a sample size of $n = 3$ animals per group. For assessment of mean septal thickness, an increase of 2 μm in mean septal thickness was considered relevant, resulting in an effect size of $d = 11.11$ (using the hyperoxia-treated wild-type mouse lungs as reference values), and using $\alpha = 0.5$ and a power (1-$\beta$) of 0.8, required a sample size of $n = 2$ animals per group. Since both the total number of alveoli in the lung and the mean septal thickness are measured in the same animals, a sample size of $n = 3$ animals per group was required, which was extended to four animals per group, in the event of an outlier arising from technical issues related to tissue processing during embedding for stereological analysis.

Data are presented as mean ± SD. Differences between groups were evaluated by one-way ANOVA with Tukey's *post hoc* test for multiple (more than two) comparisons, while two-group comparisons were performed with an unpaired Student's *t*-test. All statistical analyses were performed with GraphPad Prism 6.0. For microarray studies, a Welch's approximate *t*-test was used to determine *P* values which were corrected using the algorithm of Benjamini and Hochberg, to generate the corrected *P*-value, *P*(corr) (Benjamini & Hochberg, 1995). The presence of statistical outliers was tested by Grubbs' test, and no outliers were found. In general, data sets were too small to test normal distribution, and normality was assumed.

# Data Availability

Microarray data comparing microRNA steady-state levels in lungs of mouse pups exposed to 21% $O_2$ *versus* 85% $O_2$ are available at the GEO database under accession number GSE89666 (https://www.ncbi.nlm.nih.gov/geo/query/acc.cgi?acc = GSE89666).

**The paper explained**

**Problem**

Bronchopulmonary dysplasia (BPD) is a common and severe complication of preterm birth, where the lungs of preterm born infants do not properly develop. Notably, the formation of the alveoli—the principal gas-exchange units of the lung—is stunted, which has important consequences for the long-term respiratory health of BPD survivors. While oxygen support of infants with acute respiratory failure causes BPD, the disease mechanisms that underlie the stunted lung development are unknown.

**Results**

Our report identifies the interaction between microRNA-34a and the mRNA encoding platelet-derived growth factor receptor (PDGFR)α as a disease-relevant interaction in stunted lung developed associated with BPD that was experimentally modeled in mice. Our report also documents that this interaction is "druggable", and can be manipulated *in vivo* to protect the development of alveoli from the damaging effects of oxygen support.

**Impact**

Our report highlights a new pathological pathway that can also be pharmacologically targeted to attenuate experimental disease pathology. Targeting this specific microRNA-34a/mRNA interaction may form the basis of a new approach to the medical management of BPD.

Microarray data comparing mRNA steady-state levels in lungs of antimiR-treated mouse pups exposed hyperoxia are available at the GEO database under accession number GSE89730 (https://www.ncbi.nlm.nih.gov/geo/query/acc.cgi?acc = GSE89730).

**Expanded View** for this article is available online.

# Acknowledgements

The authors acknowledge the assistance of Ewa Bieneck (University of Giessen School of Medicine) with histological preparations, Luciana C. Mazzocchi (University of Giessen School of Medicine) for expert advice, and Ann Atzberger (Max Planck Institute for Heart and Lung Research) for expert assistance with flow cytometry. This study was supported by the Max Planck Society; Rhön Klinikum AG grant FI_66; University Hospital Giessen and Marburg grant UKGM62589135; the Federal Ministry of Higher Education, Research and the Arts of the State of Hessen "*LOEWE* Programme", the German Center for Lung Research (*Deutsches Zentrum für Lungenforschung*), and by the German Research Foundation (*Deutsche Forschungsgemeinschaft*) through Excellence Cluster EXC147, Collaborative Research Center SFB1213/1 (project A03), Clinical Research Unit KFO309/1 (projects P2, P5, P6, P8, and Z1), and individual research grant Mo 1789/1.

# Author contributions

JR-C performed transgenic animal, and *in vivo* target site blocker, mimic, and antimiR studies. JR-C, EL, and CN performed *in vitro* hyperoxia, target site blocker, mimic, and antimiR studies. JR-C, EL, and CN performed the stereology analyses. JR-C, JQ, FP, and SH performed flow cytometry studies. EL and ES performed and analyzed the microarray studies and performed bioinformatics analyses. DESS performed cryosection immunofluorescence studies. PFA performed selected cell-culture experiments. IM, IV, JAR-C, and KA assisted with transgenic animal, target site blocker, and antimiR animal experiments. JR-C, JQ, ES, CN, IV, JAR-C, WDR, KA, SH, WS, and REM conceived experiments, analyzed data, supervised experiments, and

provided essential reagents, equipment, and infrastructure. JR-C, WS, and REM conceived the study, directed the study, and wrote the manuscript.

## Conflict of interest

The authors declare that they have no conflict of interest.

## For more information

(i) The American Lung Association (English Language) information page on bronchopulmonary dysplasia: https://www.lung.org/lung-health-and-diseases/lung-disease-lookup/bronchopulmonary-dysplasia/

(ii) The British Lung Foundation (English Language) information page on bronchopulmonary dysplasia: https://www.blf.org.uk/support-for-you/support-for-you/bronchopulmonary-dysplasia-bpd/what-is-it

(iii) The KidsHealth patient (English Language) information page for parents of infants with bronchopulmonary dysplasia: https://kidshealth.org/en/parents/bpd.html

(iv) The (German Language) information page of the Federal Association "The Preterm Infant": https://www.fruehgeborene.de/

(v) The microRNA database: http://www.mirbase.org/

(vi) The Mouse Genome Informatics international database resource for the laboratory mouse: http://www.informatics.jax.org/

(vii) GenBank human miR-34a entry: https://www.ncbi.nlm.nih.gov/gene/407040

(viii) Genbank mouse miR-34a entry : https://www.ncbi.nlm.nih.gov/gene/723848

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
