## [Review Process File · EMBO Molecular Medicine]

Targeting miR-34a/Pdgfra interactions partially corrects alveologenesis in experimental bronchopulmonary dysplasia

Jordi Ruiz-Camp, Jennifer Quantius, Ettore Lignelli, Philipp F. Arndt, Francesco Palumbo, Claudio Nardiello, David E. Surate Solaligue, Elpidoforos Sakkas, Ivana Mižková, José Alberto Rodríguez-Castillo, István Vadász, William D. Richardson, Katrin Ahlbrecht, Susanne Herold, Werner Seeger, and Rory E. Morty

Review timeline:

Submission date:	16 June 2018
Editorial Decision:	20 June 2018
Authors' Appeal:	20 June 2018
Editorial Decision:	13 September 2018
Revision received:	12 December 2018
Editorial Decision:	14 January 2019
Revision received:	19 January 2019
Accepted:	23 January 2019

Editor: Lise Roth

Transaction Report:

1st Editorial Decision

20 June 2018

Thank you for submitting your manuscript to EMBO Molecular Medicine. I have now had a chance to read your research article carefully and to discuss it with the other members of our editorial team. I am sorry to inform you that we find that the manuscript is not well suited for publication in EMBO Molecular Medicine and that we therefore have decided not to proceed with peer review.

Your study investigates interactions between miRNAs and PDGFR α in the lung alveolarization process in the context of bronchopulmonary dysplasia (BPD) in neonates. A miRNA microarray performed in a BPD model identified that miR-34a was upregulated in hyperoxia-exposed lungs. Global miR-34a deletion, or PDGFR α -positive cells specific miR-34a deletion protected against hypoxia-driven arrest of alveolarization. Disrupting the interaction mir-34a/Pdgfra 3'UTR had similarly a beneficial effect. Finally, therapeutic application of antimiR-34a protected alveolarization from hyperoxia, increased the number of PDGFR α positive cells, and lung myofibroblasts were partially restored.

We recognize that your findings suggest a potentially druggable pathway in the context of hyperoxia-driven arrested lung alveolarization following pre-term birth, and we really appreciate the completeness of your experimental approach. However, a previous study reporting the therapeutic targeting of miR-34a in epithelial cells in the context of PBD detracts from the kind of conceptual advance we expect from an EMBO Molecular Medicine article. Therefore, I am afraid that we cannot offer further consideration to your article.

That being said, I discussed your work with Dr. Andrea Leibfried, executive editor of the new open-access journal Life Science Alliance. Life Science Alliance is launched as a partnership between EMBO Press, Rockefeller Press, and Cold Spring Harbor Laboratory Press, and publishes work that is of high value to the respective communities across all areas in the life sciences. I am glad to say that Andrea is in principle interested in publication of the manuscript at Life Science Alliance and

she would be pleased to send your manuscript in its current form out for formal peer-review. This would require you to transfer the paper to Life Science Alliance via the link below. No reformatting is required.

I am sorry that we cannot be more positive on this occasion.

Authors' Appeal

20 June 2018

Authors to EMBO Molecular Medicine editor:

[...] Your very nicely worded (thank you!) letter below effectively summarised our submission letter and the Abstract of our submission. You have cited lack of conceptual novelty as the key reason for not reviewing our manuscript, citing another recent manuscript that has addressed miR.34a in experimental bronchopulmonary dysplasia (BPD), albeit in a different system, different pathway, and different cell-type.

In your summary of your reading of our manuscript, and your discussions with other experts, you appear to have not noticed the considerable conceptual (and novel) advance, represented by being able to alter the pathological course of a disease by interfering specifically with a microRNA interaction with a single specific mRNA target. This is entirely novel, in any context; but appears to have been completely missed by you and your colleagues. There are different ways to look at conceptual advance. Finding a "new player" in a process is one, but identifying an entirely new way to interfere with a pathological process is also a (major) conceptual advance; and given the emerging interest of microRNA as players in organ development and disease pathogenesis, this represents a very new and specific manner of modulating microRNA-directed processes, that goes way beyond the microRNA inhibitor (antimiR) approach (which we also undertook by way of comparison), and which is of very broad interest in the arena of molecular medicine. [...]

EMBO Molecular Medicine chief editor's response:

[...] In your case, despite the interest of the study, and as we said already, the recent publication of a paper on the therapeutic targeting of miR-34a in BPD, even in a different system, different pathway, and different cell-type, was found to compromise the conceptual advance of the findings; indeed EMM focuses on those studies that provide significant novel insights of a clinical and/or translational nature. The fact that targeting miR34a was found already beneficial in BPD is compromising the conceptual advance of your findings, even if your study provides novel insights such as interfering with microRNA/mRNA interaction. Further, this study was not discussed in your manuscript, which participated in underlining the limitations of your work. We cannot of course exclude having overlooked the specific advance represented by using TSB technology in vivo presented in your study, and therefore, we would be ready to re-evaluate your manuscript. If you would like our editors to re-evaluate your manuscript, we would please ask you to formulate an appeal letter based strictly on the scientific aspects and conceptual advance of your work that you believe we may have misunderstood.

Authors' response:

[...] considering the novelty and scientific substance of our manuscript, a very key issue in your lack of interest in our study is the apparent lack of novelty of our, considering the "other" recently published manuscript. If these two manuscripts are considered side-by-side, there is actually very little overlap, apart from consideration of miR-34a in the context of alveolar development. The already-published manuscript addressed miR-34a in the epithelium, and utilised an antimiR-based approach to "therapeutically" manage experimental bronchopulmonary dysplasia (BPD). That latter point is the only area of overlap of the two manuscripts. Our manuscript utilised the same antimiR-based approach, to make the same point: miR-34a is important in aberrant lung alveolarization. In this age of non-reproducible results, this validation is a strength of our study, not a weakness. However, the core of our study was the function of miR-34a in the mesenchyme (not at all considered in the already-published manuscript). In our study, we make entirely novel observations

about the role of miR-34a in alveolar myofibroblasts, using a complex transgenic mouse created entirely for this purpose. After creating this very solid base, then we then proceeded to do something conceptually very novel: we identified the pathologically-relevant microRNA target in the relevant lung compartment, and then selectively interfere with this microRNA-mRNA interaction in vivo, in an experimental animal model, and demonstrate the therapeutic utility of target-site blocker technology in the management of experimental disease. That approach worked very well, was absolutely conceptually novel, and is of very board interest, since it can be expanded to any organ and any organisms and any disease model. Furthermore, our study was conducted to the highest technical standards, using design-based stereology to robustly analyse and quantify lung structure. None of that can be said for the already-published study. Additionally, our study considered two separate-but-related events in lung development: first, the production of nascent alveoli by secondary septation in pre-existing alveoli; and second, but equally important, the progressive thinning of the alveolar walls as lung development proceeds. These two events, which are difficult to dissect apart, were not even considered in the already-published paper. To specialists in the field, the two manuscripts (our, and the study already published) are actually wonderfully complementary in what they report. The already-published study reports limited partial restoration of lung development in the disease model, addressing miR-34a in the lung epithelium. Our study reports substantial partial restoration of lung development in the disease model, addressing miR-34a in the lung mesenchyme. Together, between the two papers, we probably have the whole miR-34a story! However, our paper also moves beyond that complementarity by describing conceptually novel approaches to the management of lung (and perhaps other) disease, by interrupting microRNA-mRNA interaction (thus, having a specific mRNA target using a target-site blocker, as opposed to influencing hundreds of mRNA targets using the antimiR approach). If you like, I can summarize the strengths of our study here:

- (1) Validation of the antimiR studies in the already-published paper
- (2) Consideration of miR-34a function in the lung mesenchyme (entirely novel).
- (3) Consideration of miR-34a function in two separate lung development processes: alveolarization and septal thinning (not considered before, anywhere).
- (4) Consideration of target-site blocker technology which (i) identified a pathologically-relevant microRNA-mRNA interaction (conceptually entirely novel) and (ii) demonstrated the utility of target-site blocker technology to interrupt a pathologically-relevant microRNA-mRNA interaction to manage lung disease in an animal model (conceptually absolutely novel, and of very broad interest).
- (5) Experiments conducted to the highest technical standard, utilizing design-based stereology of quantify development of the lung structure.
- (6) is highly complementary to the already-published paper, where the two papers together probably tell the full miR-34a/lung development story in BPD.

[...] In light of what I have outlined above in response to your fourth point, concerning the scientific aspects of our study: yes, I would like the Editors to reconsider the decision to triage our study. [...]

I thank you for your email, and your considered thoughts, and I hope that you will take the points that I have raised above in the constructive spirit in which they are intended.

2nd Editorial Decision

13 September 2018

Thank you for the submission of your manuscript to EMBO Molecular Medicine. We have now heard back from the referees who were asked to evaluate your manuscript.

As you will see from the reports below, while referees 1 and 2 are positive and support publication of the article in EMBO Molecular Medicine pending appropriate revisions, referees 3 and 4 feel that the claims are overstated, and in particular referee 3 is not convinced that the conclusions on lungs differences are supported by the data. Moreover, referee 3 points to a potential conceptual flaw that must be addressed satisfactorily. This referee is puzzled by the fact that cells sorted according to PDGFR α expression express the most miR-34a, which would suggest that miR-34a expression has little to do with PDGFR α expression in these cells.

Addressing the reviewers' concerns in full will be necessary for further considering the manuscript in our journal. Particular attention should be given to rewriting the manuscript and tuning-down some of the claims as largely suggested by referees 3 and 4. EMBO Molecular Medicine encourages

a single round of revision only and therefore, acceptance or rejection of the manuscript will depend on the completeness of your responses included in the next, final version of the manuscript.

Please also contact us as soon as possible if similar work is published elsewhere. If other work is published, we may not be able to extend the revision period beyond three months.

I look forward to receiving your revised manuscript.

***** Reviewer's comments *****

Referee #1 (Remarks for Author):

This is a very interesting study that may have identified a new an important new target that inhibits alveolarization in blouse in represent to hyperopia. It involves interaction between a specific microSNA and PDGF signaling. This is potentially therapeutically tractable.

I only have two relatively minor issues.

The n of 4 is justified by sample size calculations but still strikes me as on the small side to make a valid statistical comparison.

The question of whether this finding may be relevant or not in humans is not discussed and should be since findings in mice are not always well replicated in human studies. This is important because the authors are not really interested in finding a therapy for alveolar dysplasia in mice but BPD in humans.

Referee #2 (Remarks for Author):

The manuscript "Targeting miR-34a/Pdgfra interactions corrects alveologenesi in experimental bronchopulmonary dysplasia" details extensive and elegant studies to establish the interaction between miR-34a and PDGFRa. This group are the first to establish this type of interaction and to offer modulation and miR34a as a potential target for therapeutic development. The studies are well designed and described and validated the authors findings.

There one minor suggestion:

1) While suppression of miR-34a increased the levels of PDGFRa is it unclear whether this observation was due to an increase in myofibroblast number or an increased expression of PDGFRa on existing cells.

If it is an increase in cell number, what is the mechanism by which these cells are decreased? Apoptosis? And how does miR-34a protect them or induce proliferation of this particular cell?

Referee #3 (Comments on Novelty/Model System for Author):

Specific targets to enhance abnormal lung development after neonatal injury are needed. The authors use robust methods to test specific molecular pathways

Referee #3 (Remarks for Author):

Ruiz-Camp and colleagues present their work testing the hypothesis that hyperopia induced pulmonary miR-34a expression disrupts Pdgfra expression and contributes to abnormal lung development. These data are clearly presented. Robust interrogations are performed, and multiple

different in vitro and in vivo approaches are employed. The reader benefits from a very comprehensive data presentation in the form of figures, supplemental tables and supplemental figures. This represents an incredible amount of work.

However, having read the manuscript, I am not convinced that these data support the authors conclusions. It seems that in their various approaches to manipulate miR-34a expression, the lungs are more similar than different when compared to similarly exposed WT mice. Various measures are used and presented in the tables (S1, S3, S4, S5) and the differences in stereological analysis show that the attenuation of lung injury is not uniform. Furthermore, more indices of development are NOT different b/w WT and "miR-34a altered" than are different, proving that the lungs are likely less protected than the authors conclusions. If there is a difference, it appears to be in septal thickness. That finding appears consistent and robust. More focus should be spent on this finding, if not experimentally, than in how these data are presented and interpreted.

Is it not counterintuitive that cells sorted on PDGFRa expression, express the most miR-34a? Wouldn't this suggest that miR-34a expression has little to do with PDGFRa expression in these cells?

Did the authors consider other targets of miR-34a and whether or not these targets could be implicated in lung injury?

Minor:

- 1) Text Has figure 2D labeled as cells and mice.
- 2) Some experiments could benefit from clarification on replicates, and the number of times repeated (ie, when just WB are shown. If densitometry is not presented, at least legends should state the number of times the experiment was repeated.
- 3) could the authors show that the TSB1 and 2 don't target SIRT1 miRNA via a similar sequence?

Referee #4 (Remarks for Author):

Ruiz-Camp et al. describes a potential and interesting pathway for future treatment of BPD, by targeting mir34a. The authors use the hyperoxia model to induce BPD-like symptoms in mice, and thereafter use multiple ways to decrease the levels of mir34a. The different models show significant increases in number of alveoli, but not always in the septal thickness. I think that this is a paper that should be published, but it needs a bit of correction to the text. For example, I think that the authors from time to time use too strong words to describe the improvement in alveologenesi. They are not always that dramatic.

Specific comments:

Introduction:

Is it confirmed that elastin cables (and not the upregulation of a-sma) drive the secondary septation?

To me it sounds excessive to write that the alveoli numbers are partly normalized in mir34^{-/-} mice (Fig.1c). Even though there is a one*-significant increase in mir34a^{-/-} compared to wt after hyperoxia, there is still a huge effect on the numbers of alveoli compared to normoxia.

Comment why the septal thickness decreases to less than normoxic conditions when inhibiting mir34a both genetically and with inhibitors (Fig. 1D and 4E). Is the reduction significantly compared to WT? Isn't such a reduction also detrimental for the lung?

Comparing Fig.1C and 1F:

Why is the wt decrease in number of alveoli **** in Fig.1C and only * in Fig.1F? They look very similar in the graph.

Fig.1G

What is the reason to reduced septal wall thickness in Mir34bc^{-/-} after hyperoxia?

Comparing Fig.1A with 2C:

How come that lung fibroblasts in vitro respond with increase in gene expression of all mir34-a/b/c, when only mir34-a increases in vivo? Please, comment.

Page 5, last three rows:

Fig.2D is referred to twice in the text, both as cells and as mice. Reference to Fig.2E is missing.

Page 6:

The authors comment to why mir-34a deletion in Pdgfra⁺ cells have no effect on the septal thickness, but the reference that follows (Nardiello 2017) does not explain anything about any Miglyol/tamoxifen solvent effects.

Are there other possible explanations to why the septal thickening is not improved by mir-34a deletion? Is it proved that the septal thickening and decrease in alveoli number always go hand-in-hand, or could it be two different processes- dependent on different signaling pathways?

Fig.2I-K - How does the lung histology look in mutant mice exposed to normoxia?

Fig.3B - there is a lot of background on the blot for SIRT1, it is not suitable for quantifications.

Fig.3C - from the normalized data it seems as TBS2 alone (lane 5) would have more (or at least as much) effect as TBS1+TBS2. Comment?

Fig.3D - the in vivo effect of TBS1+2 is very limited. I do not agree with the authors that there is a substantial protection (page 7). There is for example no effect on the MLI (table S4).

How was asma and pdgfra + cells quantified? Please, add a representative image as supplemental.

Table S3 - Is the genotype of mice to the right in the table written wrong? If the mice were treated with tamoxifen they should be denoted miR-34a^{i(Δ)PC/i(Δ)PC} and not mir-34a fl/fl, right?

Table S5 - how come the MLI does not improve after antimir34a during hyperoxia? Even though there is no significant difference, the trend instead suggests that the MLI gets worse. Please, comment.

Page 17 - explain/add reference to G*Power 3.1.9.2.

There is a protocol for primary lung fibroblasts and culture, where were these primary cell cultures used? The results mentioned seems to all come from the cell line MLg.

2nd Revision - authors' response

12 December 2018

(see next page)

POINT-BY-POINT REBUTTAL

Comments from the Editor:

C1: *As you will see from the reports below, while referees 1 and 2 are positive and support publication of the article in EMBO Molecular Medicine pending appropriate revisions, referees 3 and 4 feel that the claims are overstated, and in particular referee 3 is not convinced that the conclusions on lungs differences are supported by the data. Moreover, referee 3 points to a potential conceptual flaw that must be addressed satisfactorily. This referee is puzzled by the fact that cells sorted according to PDGFR α expression express the most miR-34a, which would suggest that miR-34a expression has little to do with PDGFR α expression in these cells.*

Addressing the reviewers' concerns in full will be necessary for further considering the manuscript in our journal. Particular attention should be given to rewriting the manuscript and tuning-down some of the claims as largely suggested by referees 3 and 4. EMBO Molecular Medicine encourages a single round of revision only and therefore, acceptance or rejection of the manuscript will depend on the completeness of your responses included in the next, final version of the manuscript.

R1: Thank you for your summary of the reviewer's comments. You have highlighted three specific issues: **(i)** In our manuscript, as clarified in the responses to comments C8 (Reviewer #3) and C14 (reviewer #4), we have toned down our assessment of the impact of the interventions that we report, which Reviewer #3 and Reviewer #4 felt were overstated. We have also toned down the title of our manuscript. **(ii)** We have clarified for Reviewer #3 how we believe our data support our conclusions (which have now, in some instances, been toned down). These changes are also explained in the response to comment C8, below. **(iii)** As you have mentioned, Reviewer #3 raised concerns about a possible conceptual flaw related to the detection of miR-34a in PDGFR α ⁺ cells, which perhaps reflects the complexity of some of our argumentation. We do not believe that there is any conceptual flaw, and have gone to considerable lengths to clarify this in the manuscript, and we have provided some additional original data plots (Appendix Figure S13D), additional discussion (p. 9, para. 2) in the manuscript, as well as in this rebuttal letter below (in the response to comment C9), which we hope satisfactorily addresses this issue.

Comments from the Reviewers:

Referee #1 (Remarks for Author):

C2 (General): *This is a very interesting study that may have identified a new an important new target that inhibits alveolarization in blouse in represent to hyperopia. It involves interaction between a specific microSNA and PDGF signaling. This is potentially therapeutically tractable. I only have two relatively minor issues.*

R2 (General): Thank you for your positive assessment of our study.

C3: *The n of 4 is justified by sample size calculations but still strikes me as on the small side to make a valid statistical comparison.*

R3: Thank you for the comment. We had initially included n numbers that reflected the power analysis. We have addressed the concerns of the reviewer as follows: **(i)** The data panels in Fig. 1A, and Appendix Fig. S1B formerly contained a range of n between 4 and 6. We have now repeated this experiment, and consistently report an $n=6$ for all conditions. Given that the first author (JRC) has now left our laboratory, a new set of data was generated by author EL. Given the experiment-to-experiment variance of original ΔCt values, the two sets of ΔCt values were not combined to create a larger group, as this generated a spread of data that incorrectly represented the reality of data spread in a single experiment. Rather, we have entirely replaced the six data panels, with newly-generated data that consistently report $n = 6$ (animals) per group. This new data set perfectly replicates the trends observed in the original data set. **(ii)** We have stereologically assessed additional mouse samples of lung tissue that were generated in the original experiments (but not counted), and thereby contributed additional experimental animals to now consistently report $n=5$ animals per group for the data-sets presented in Fig. 1C, D, F, G; Fig. 2J, K. This new data sets perfectly replicate the trends observed in the original data sets. The inclusion of additional animals into each of these data sets have changed every mean value and every P -value presented in Appendix Tables S1, S2, and S3; and Figures Fig. 1C, D, F, G; Fig. 2J, K. However, the trends reported for the $n=5$ exactly replicate the trends reported in the original ($n=4$) datasets. Clearly, additional n 's can still be added if need be.

For the analysis of miR-34a levels in flow-sorted PDGFR α^+ cells from P5 mouse lungs, the original plot contained data from four mice. Given that the first author (JRC) has now left our laboratory, a new set of data was generated by author FP. Again, given the experiment-to-experiment variance of original ΔCt values, the two sets of ΔCt values were not combined to create a larger group, as this generated a spread of data that incorrectly represented the reality of data spread in a single experiment. Rather, we have included the second independent confirmation as Appendix Fig. S6. This new data set perfectly replicates the trends observed in the original data set.

For the estimation of the number of PDGFR α^+ cells (Fig. 3G) and $\alpha SMA^+/PDGFR\alpha^+$ cells (Fig. 3H) after target-site blocker treatment (formerly $n=4$) has now been entirely re-

peated with an $n=5$ animals per group, and these new data replace the original data set. This new data set perfectly replicates the trends observed in the original data set.

We hope that the expansion of the n numbers in these experiments addresses the concerns of this Reviewer.

C4: *The question of whether this finding may be relevant or not in humans is not discussed and should be since findings in mice are not always well replicated in human studies. This is important because the authors are not really interested in finding a therapy for alveolar dysplasia in mice but BPD in humans.*

R4: Thank you for this important comment related to the translational extension of the work. This is an important point. Shortly prior to the submission of our original manuscript, a report appeared (Syed *et al.*, 2018, in our reference list) that reported increased miR-34a levels in the lungs of human infants with bronchopulmonary dysplasia. I believe that this exactly answer the question of the reviewer. Along these lines, this finding is briefly discussed in our manuscript (p. 11, para. 1).

Referee #2 (Remarks for Author):

C5 (General): *The manuscript "Targeting miR-34a/Pdgfra interactions corrects alveologenesis in experimental bronchopulmonary dysplasia" details extensive and elegant studies to establish the interaction between miR-34a and PDGFRa. This group are the first to establish this type of interaction and to offer modulation and miR34a as a potential target for therapeutic development. The studies are well designed and described and validated the authors findings. There one minor suggestion:*

R5 (General): Thank you for your very positive comments about our manuscript.

C6: *1) While suppression of miR-34a increased the levels of PDGFRa is it unclear whether this observation was due to an increase in myofibroblast number or an increased expression of PDGFRa on existing cells. If it is an increase in cell number, what is the mechanism by which these cells are decreased? Apoptosis? And how does miR-34a protect them or induce proliferation of this particular cell?*

R6: Thank you for this important comment, which also pertains to questions raised by Reviewer #3. We have performed a number of new *in vivo* and *in vitro* studies to address this point. Concerning apoptosis and proliferation, we have first assessed the apoptosis status of PDGFR α ⁺ cells in the lungs of P5 mice exposed to normoxia and hyperoxia using annexin V-based flow cytometry, to answer whether hyperoxia exposure may decrease PDGFR α ⁺ cell abundance. The answer is yes (Appendix Fig. S13). The number of PDGFR α ⁺ cells recovered from P5 mouse pups was too low to generate a meaningful S-phase analysis by flow cytometry (the G0/G1 and G2/M peaks are not well resolved in the histogram in Appendix Fig. S14). For this reason, we turned to Ki67 labelling in cryosections from mice expressing nuclear-localised GFP from the *Pdgfra* promoter, allowing us to score PDGFR α ⁺ cells as Ki67⁺ (thus, proliferating). These data revealed that hyperoxia exposure decreases the number of PDGFR α ⁺ cells that proliferate in P5 mouse lungs (Appendix Fig. S15-S18). These studies answer the question of whether hyperoxia can decrease PDGFR α ⁺ cell abundance by impacting apoptosis and proliferation *in vivo*: it can. In parallel, to further address the concerns of the Reviewer, we have also performed new *in vitro* studies, where the miR-34a mimic was transfected into primary mouse lung fibroblasts, where no impact on baseline apoptosis was noted, although a slight negative impact on proliferation was noted (Appendix Fig. 19). Collectively, these data indicate that hyperoxia increases apoptosis and decreases proliferation of PDGFR α ⁺ cells *in vivo* in developing mouse lungs, and that *in vitro*, miR-34a impacts serum-stimulated proliferation, but not baseline apoptosis, of primary mouse lung fibroblasts, used as a surrogate for PDGFR α myofibroblasts.

Turning to the question of whether the abundance of PDGFR α is impacted within the PDGFR α ⁺ cells: hyperoxia reduced the abundance of PDGFR α on individual PDGFR α ⁺ cells. This is evident from the newly-included, original flow cytometry scattergrams (which are also transformed to zebra plots to delineate cell density relative to fluorescence intensity) presented in Appendix Fig. S13D. These are repeated below and annotated for ease of reference:

It is clear from the PDGFR α^+ gating, that under normoxic conditions, PDGFR α^+ cells span the north-south spectrum of the gate. Under hyperoxic conditions, PDGFR α^+ cells have been lost from the upper half of the gate. As this is a log-scale, cells that would occupy the upper half of the gate are the most highly fluorescent of the PDGFR α^+ cell population (thus, express the most PDGFR α per cell). These data demonstrate that hyperoxia exposure caused loss of fluorescence (thus, PDGFR α abundance) in PDGFR α^+ cells. It is important to note that this impacts our interpretation of the (a) abundance of PDGFR α on PDGFR α^+ cells, and (b) the total number of PDGFR α^+ cells assessed, since when the PDGFR α^+ “fluorescence” falls beyond that defined by the threshold set by the lower (south) border of the PDGFR α^+ cell gate, these “fluorescence low” cells will now be scored as PDGFR α -negative, and hence, decrease the PDGFR α^+ cell number assessed.

We believe that the hyperoxia effect is mediated by miR-34a, since hyperoxia drove increased miR-34a levels in PDGFR α^+ cells flow-sorted from the lungs of hyperoxia-exposed mice (Fig. 2F); and by either increasing miR-34a levels with a mimic (Fig. 2B) or neutralizing miR-34a with an anti-miR-34a (Fig. 2E) or antagonising the miR-34a/*Pdgfra* interaction (with target-site blockers; Fig 3B,C) the corresponding effect on PDGFR α levels were noted. Indeed, in our new data included as Fig. 2E, the presence of an anti-miR-34a partially protected PDGFR α levels from the impact of hyperoxia. We hope that these new data, and the interpretation of these data provided here, and in the manuscript address the concerns of the reviewer.

Referee #3 (Comments on Novelty/Model System for Author):

C7 (General): *Specific targets to enhance abnormal lung development after neonatal injury are needed. The authors use robust methods to test specific molecular pathways*

R7 (General): Thank you for your positive assessment of our study.

Referee #3 (Remarks for Author):

C8 (General): *Ruiz-Camp and colleagues present their work testing the hypothesis that hyperopia induced pulmonary miR-34a expression disrupts Pdgfra expression and contributes to abnormal lung development. These data are clearly presented. Robust interrogations are performed, and multiple different in vitro and in vivo approaches are employed. The reader benefits from a very comprehensive data presentation in the form of figures, supplemental tables and supplemental figures. This represents an incredible amount of work.*

However, having read the manuscript, I am not convinced that these data support the authors conclusions. It seems that in their various approaches to manipulate miR-34a expression, the lungs are more similar than different when compared to similarly exposed WT mice. Various measures are used and presented in the tables (S1, S3, S4, S5) and the differences in stereological analysis show that the attenuation of lung injury is not uniform. Furthermore, more indices of development are NOT different b/w WT and "miR-34a altered" than are different, proving that the lungs are likely less protected than the authors conclusions. If there is a difference, it appears to be in septal thickness. That finding appears consistent and robust. More focus should be spent on this finding, if not experimentally, than in how these data are presented and interpreted.

R8 (General): Thank you for highlighting the strengths of our study, and also for raising some important issues for clarification. To the specific concerns raised in the second paragraph of the Reviewer's comment:

(i) The Reviewer states that, when using various interventions to interrupt miR-34a, the lungs are more similar than different to similarly injured intervention-control mice. This implies that the intervention lungs are still closer in structure to

“sick” lungs than they are to healthy lungs. We do not entirely agree with this assessment, as outlined in the table below:

Intervention	Impact of intervention on secondary septation (generation of new alveoli) under hyperoxic conditions	Impact of intervention on septal thinning (maturation of septal thickness) under hyperoxic conditions
miR-34a knockout	34% “increase”	Septal thickness “beyond normalized”. The mean value was even thinner than healthy, wild-type mice.
Knockout of miR-34a in PDGFR α ⁺ cells	42% “increase” (almost a double)	Not relevant*
Disruption of miR-34a/ Pdgfra interaction (target-site blocker)	25% “increase”	Septal thickness normalized.
Neutralisation of miR-34 with anti-miR-34a	40% “increase” (almost a double)	Septal thickness normalised.

*Septal thickening did not occur in response to hyperoxia, as is well-known when tamoxifen solvents such as cottonseed oil, which is chemically similar to Miglyol, are applied to newborn mouse pups.

In two out of four instances, the septal thickness was truly normalized. In another instance, the septal thickness was reduced to below that noted in normoxia-exposed, wild-type mice. So, in terms of septal thickness, the “treated” lungs are *more different than similar* to the control-intervention injured lungs (the opposite of what the reviewer has stated). In terms of alveoli number, the reviewer is correct; the “treated” lungs remain more similar than different to the control-intervention injured lungs, because they did not cross a 50% threshold between healthy and sick lungs. This is unsurprising. Arrested alveolar development in response to hyperoxic insult is multifactorial: changes in gene expression (in our study, our microarray documents changes in the abundance of over 1200 mRNA species at P5), including microRNA expression, are impacted, but perturbations to extracellular matrix production and processing, as well as inflammation, notably mediated by residential alveolar macrophages, also play a role. By targeting a single pathogenic factor (miR-34a) we target only a part of the pathological pro-

cess and as such, we can only reasonably expect a partial recovery of the phenotype, which is indeed what we see.

(ii) To the specific points raised by the reviewer here: we agree! But, we feel that even the moderate improvement in alveolarization is noteworthy and an important foundation for more work in this area. In terms of action to the particular critique “*proving that the lungs are likely less protected than the authors conclusions*”, we have toned down our conclusions, where you have correctly noted that we have – at times – overstated the impact of the intervention in our discussions. We have also toned down the title of our article.

(iii) In the second point of the Reviewer, the Reviewer is correct that the attenuation of the lung injury using different interventions is not uniform (Appendix tables S1, S3, S4, and S5). We would not expect the attenuation of the effects to be uniform, because each effect either targets a different cell, or a different microRNA-mediated process(es). For example: **(a)** A global knockout of miR-34a will impair miR-34a function in every cell, and impact every miR-34a target (keeping in mind that miR-34b and miR-34c are still present). **(b)** An inducible knockout of miR-34a in PDGFR α^+ cells will be much more restricted in its impact, since the impact is restricted to cells where the *Pdgfra* promoter is operative, and only after the induction of the ablation. In spite of that, the largest effect in alveolarization was noted (42% increase), highlighting the importance of miR-34a in PDGFR α^+ cells in aberrant alveolarization **(c)** Disrupting the miR-34a/*Pdgfra* interaction will impact every cell-type, but the impact will be limited to the consequences of disrupting the miR-34a/*Pdgfra* interaction, and all other miR-34a interactions with other miR-34a mRNA transcript targets will be unaffected. Despite the very specific nature of the intervention, the intervention resulted in a 25% increase in alveoli number, highlighting the importance of this very specific intervention in aberrant lung alveolarization. **(d)** General neutralisation of miR-34a activity with an anti-miR-34a will neutralize the effects of miR-34a in every cell-type, and will impact every miR-34a mRNA transcript target; hence, the effect here is anticipated to be amongst the most impactful, which is indeed the case.

(iv) We have focused on the alveoli number in our study, since this is the currently intractable clinical issue with infants with BPD: too few alveoli. While the increase in the number of alveoli is moderate (between 34% and 42%), it is tremendously exciting and important, since being able to restore that number of alveoli in severe BPD patients, or indeed, COPD patients, would be a life-changing event on its own. We feel

that our observation generates a proof-of-principle studies that will – in time – be expanded upon to further increase the restoration of alveoli, perhaps with a combination of interventions that include antimiR-34a or the miR-34a/*Pdgfra* target-site blockers. To address the concern of the reviewer “*it appears to be in septal thickness. That finding appears consistent and robust. More focus should be spent on this finding, if not experimentally, than in how these data are presented and interpreted*” we have expanded our discussion on the implications of the septal thickness findings (p. 10, end of para. 2.).

C9: *Is it not counterintuitive that cells sorted on PDGFR α expression, express the most miR-34a? Wouldn't this suggest that miR-34a expression has little to do with PDGFR α expression in these cells?*

R9: Thank you for your question, which was also raised in part by Reviewer #2. Initially, it may seem counterintuitive to look for miR-34a expression in PDGFR α ⁺ cells, since elevated miR-34a levels will reduce the abundance of PDGFR α on PDGFR α ⁺ cells, the very surface marker by which we identify and sort these cells. As we have stated in our manuscript, we believe that hyperoxia drives increased expression of miR-34a, and we report that this is a particularly prominent effect in PDGFR α ⁺ cells. At the same time, we document that hyperoxia decreases the abundance of this cell-type. We provide additional data in the new version of our manuscript that hyperoxia can increase apoptosis and decrease proliferation of PDGFR α ⁺ cells (Appendix Fig. S13-S18). In the remaining PDGFR α ⁺ cells, we believe that increased miR-34a levels will decrease the amount of PDGFR α in each individual PDGFR α ⁺ cell. This is evident from the newly-included, original flow cytometry scatter-grams (which are also transformed to zebra plots to delineate cell density relative to fluorescence intensity) presented in Appendix Fig. S13D. These are repeated below for ease of reference:

It is clear from the PDGFR α^+ gating, that under normoxic conditions, PDGFR α^+ cells span the north-south spectrum of the gate. Under hyperoxic conditions, PDGFR α^+ cells have been lost from the upper half of the gate. As this is a log-scale, cells that would occupy the upper half of the gate are the most highly fluorescent of the PDGFR α^+ cell population (thus, express the most PDGFR α per cell). These data demonstrate that hyperoxia causes loss of fluorescence (thus, PDGFR α abundance, ostensibly due to increased miR-34a expression) in PDGFR α^+ cells. It is important to note that this impacts our interpretation of the (a) abundance of PDGFR α on PDGFR α^+ cells, which we believe is decreased under hyperoxic conditions; and (b) the total number of PDGFR α^+ cells assessed, since when the PDGFR α^+ “fluorescence” falls beyond that defined by the threshold set by the lower (south) border of the PDGFR α^+ cell gate, these “fluorescence low” cells will now be scored as PDGFR α -negative, and hence, decrease the PDGFR α^+ cell number assessed. Specifically to the point that you have raised, we will definitely “lose” the ability to detect cells with the highest miR-34a levels, as the impact of comparatively high miR-34a levels on PDGFR α will cause those cells to no longer be detected. Thus, we hope that we have convinced this reviewer that it may seem counterintuitive, but it is indeed feasible to sort cells on PDGFR α expression which express high levels of miR-34a, although we do agree that we will lose the ability to detect those cells that express the highest levels of miR-34a. However, that does not change the message and conclusions of our study.

To your second point, “Wouldn't this suggest that miR-34a expression has little to do with PDGFR α expression in these cells?”, we do not believe so, since (a) *in vitro* when using the MLg cell-line as a surrogate for PDGFR α^+ cells, a miR-34a mimic de-

creased PDGFR α levels (Fig. 2B), and an anti-miR-34a increased PDGFR α levels (Fig. 2E). Similarly, (b) *in vitro*, a miR-34a/*Pdgfra* target-site blocker restored PDGFR α expression (Fig. 3B,C), and (c) *in vivo* both a miR-34a/*Pdgfra* target-site blocker (Fig. 3G) and an anti-miR-34a (Fig. 4F) increased the number of PDGFR α ⁺ cells. Thus, three lines of direct, and two lines of indirect evidence support the idea that miR-34a has a great deal to do with PDGFR α expression.

C10: *Did the authors consider other targets of miR-34a and whether or not these targets could be implicated in lung injury?*

R10: Thank you for your suggestion, which is most relevant. In response to this suggestion we have now included consideration of a third miR-34a target: c-Kit. This serves as a second control for the specificity of our target-site blocker studies (the newly included data are presented in Fig. 3B). Specifically to your second point on other miR-34a targets that may mediate injury to the developing lung in response to hyperoxia: this has very recently been addressed by another group, which identified the miR-34a target angiopoietin-1 as a causal player in arrested alveolarization (cited as Syed *et al.*, 2018; in our literature list). This has been discussed in our manuscript (p. 11, para. 1).

C11: *Minor: 1) Text Has figure 2D labeled as cells and mice.*

R11: Thank you for this comment, which we have now addressed.

C12: *2) Some experiments could benefit from clarification on replicates, and the number of times repeated (ie, when just WB are shown. If densitometry is not presented, at least legends should state the number of times the experiment was repeated.*

R12: Thank you for this comment. The number of times each experiment was repeated is now declared in the legend of every figure, for each data set (sometimes this appears in the general comments at the end of the figure legend, not at the panel description).

C13: *3) could the authors show that the TSB1 and 2 don't target SIRT1 miRNA via a similar sequence?*

R13: Thank you for this comment. The binding sites for both target-site blockers are now aligned with the 3'-UTR sequences of the *Kit* and *Sirt1* mRNA transcripts in the new Fig. 3A. A blast of the target-site blocker sequences did not reveal any other possible binding regions in either transcript.

Referee #4 (Remarks for Author):

C14 (General): *Ruiz-Camp et al. describes a potential and interesting pathway for future treatment of BPD, by targeting mir34a. The authors use the hyperoxia model to induce BPD-like symptoms in mice, and thereafter use multiple ways to decrease the levels of mir34a. The different models show significant increases in number of alveoli, but not always in the septal thickness. I think that this is a paper that should be published, but it needs a bit of correction to the text. For example, I think that the authors from time to time use too strong words to describe the improvement in alveologenesi. They are not always that dramatic.*

R14 (General): Thank you for your positive comments about our manuscript. As requested, in multiple instances, we have toned down our stronger comments about the impact of our interventions to partially correct aberrant lung alveolarization. We have similarly toned down the title of the manuscript.

C15: *Specific comments: Introduction: Is it confirmed that elastin cables (and not the upregulation of α -sma) drive the secondary septation?*

R15: Thank you for your comment. The elastin theory of alveolar development (which encompasses two different theories: the “net” and the “crest” hypotheses) suggest that the elastin cables are a key element in alveolarization. Of course, these cables are produced and remodeled by the alveolar myofibroblasts which have muscle characteristics, including the production of α SMA. Thus, I believe that the answer to the question is that both processes are relevant to alveolarization. We have included this comment in the text of the manuscript (p. 3, para 2.) which also contained citation to the Branchfield *et al.* manuscript which highlights this idea.

C16: *To me it sounds excessive to write that the alveoli numbers are partly normalized in mir34^{-/-} mice (Fig.1c). Even though there is a one*-significant increase in mir34a^{-/-} compared to wt after hyperoxia, there is still a hugh effect on the numbers of alveoli compared to normoxia.*

R16: Thank you for this comment. You are correct. We agree that the statement about partial normalization is too strong, and this has been toned down to reflect the percentage change, and the statement is no longer evaluative. Our manuscript title has also been similarly modified to town down the impact of the study.

C17: *Comment why the septal thickness decreases to less than normoxic conditions when inhibiting mir34a both genetically and with inhibitors (Fig. 1D and 4E). Is the reduction significantly compared to WT? Isn't such a reduction also detrimental for the lung?*

R17: Thank you for this question. The reduction is significant with respect to “normoxia, wild-type” and this comparison is now indicated in the artwork. We have now commented that septal thinning may occur due to matrix signaling to the epithelium, either via receptor-mediated interactions, or to matrikine gradients (p. 10, end of para. 2.). This is, however, entirely speculative, and should be read as such. You have raised an excellent point about whether this reduction is detrimental or not. We are currently developing respiratory dynamics methodologies for newborn mice, and in time, we may have an answer about how this may affect gas exchange in these animals. Up to now, all we know is that the mice are alive, and appear healthy and happy! Whether or not a breathing (or other) phenotype is associated with this septal thinning remains to be determined.

C18: *Comparing Fig.1C and 1F: Why is the wt decrease in number of alveoli **** in Fig.1C and only * in Fig.1F? They look very similar in the graph.*

R18: Thank you for noticing this mistake, which has now been corrected. Both should be ****.

C19: *Fig.1G. What is the reason to reduced septal wall thickness in Mir34bc^{-/-} after hyperoxia?*

R19: Thank you for your question. The reason is not immediately apparent. By way of suggestion, we have noted that under hyperoxic conditions, the expression of the 3p strands of miR-34b and miR-34c are both increased in abundance in the developing mouse lung (Appendix Fig. S1B). We have added this suggestion to the text of our manuscript (p. 5, para. 2).

C20: *Comparing Fig.1A with 2C: How come that lung fibroblasts in vitro respond with increase in gene expression of all mir34-a/b/c, when only mir34-a increases in vivo? Please, comment.*

R20: Thank you for the comment. We can only assume that this – at least in part – represents a difference between the *in vivo* situation, and the somewhat artificial *in vitro* situation, with MLg cells cultured on a plastic substrate. However, we would also like to point out that there is an increase in the 3p strands of miR-34b and miR-34c (Appendix Fig. S1B), suggesting some hyperoxia-responsiveness of the miR-34b and miR-34c promoter *in vivo* as well.

C21: *Page 5, last three rows: Fig.2D is referred to twice in the text, both as cells and as mice. Reference to Fig.2E is missing.*

R21: Thank you for noticing this error, which has now been corrected.

C22: *Page 6: The authors comment to why mir-34a deletion in Pdgfra⁺ cells have no effect on the septal thickness, but the reference that follows (Nardiello 2017) does not explain anything about any Miglyol/tamoxifen solvent effects.*

R22: Thank you for noticing this statement, which was improperly worded in our manuscript. The Nardiello 2017 manuscript addresses cottonseed oil, which is another widely-employed tamoxifen solvent, which protects mice against septal thickening in response to hyperoxic insult, ostensibly through providing parenteral nutrition. Miglyols are esters of saturated coconut and palm kernel oil- derived caprylic and cap-

ric fatty acids and glycerin, closely related to the mixture of saturated and unsaturated fatty acid derivatives in cottonseed oil. We believe that the protective effect of myglyol Has a similar basis to that of cottonseed oil. We have re-worded this statement in the manuscript to provide more details, and a better context (p. 6, end of para. 2).

C23: *Are there other possible explanations to why the septal thickening is not improved by mir-34a deletion? Is it proved that the septal thickening and decrease in alveoli number always go hand-in-hand, or could it be two different processes- dependent on different signaling pathways?*

R23: Thank you for your comment. Perhaps I have not understood it, but the septal thickening is significantly improved by miR-34a deletion. To your second point, septal thinning and changes in alveoli number both occur during alveolarization, and are both disturbed during aberrant lung alveolarization. While apparently going hand-in-hand, we believe them to be separate processes, with alveolarization being facilitated by subdivisions of the alveolar airspaces through the generation of secondary septa, whilst septal thinning is most likely due to the spatial rearrangement of epithelial cells that organize themselves in the newly formed septa. As this is pure speculation, we have not included this idea in the manuscript proper.

C24: *Fig.2I-K - How does the lung histology look in mutant mice exposed to normoxia?*

R24: Thank you for this comment. A representative field of normoxia-exposed mutant mice has now been included in Fig. 2I.

C25: *Fig.3B - there is a lot of background on the blot for SIRT1, it is not suitable for quantifications.*

R25: Thank you for this comment. Your comment is well taken. Our repeated stripping of the blots to get the PDGFR α , SIRT1, and β -actin generated very dirty blots. We have now redone all of our target-site blocker validation blots to address this concern. We have also changed our protocol to avoid stripping blots, but rather, we now developed blots sequentially without stripping the blot, as indicated in the uncropped blot images in Appendix Fig. S24. This yielded cleaner blots with less background, but also meant

that sometimes a “previously developed band” would generate very strong signals when a weaker band was being detected, but this was not problematic, as the affected areas of the blots did not interfere with the detection of the bands of interest. The resultant SIRT1 blots, one of which is now presented in cropped form in Fig. 3C, yields a much clearer effect, which we have not tried to quantify by densitometry. Rather, we have selected a second, independent control mRNA, that of *Kit* (encoding c-Kit), which is also targeted by miR-34a, but which – like SIRT1 – is not affected by the target-site blocker cocktail. These new data are presented in the new Fig. 3B. We believe that the combination of c-Kit and SIRT1 data make a much more robust case for the target-site blocker specificity than we had previously presented.

C26: *Fig.3C - from the normalized data it seems as TBS2 alone (lane 5) would have more (or at least as much) effect as TBS1+TBS2. Comment?*

R26: Thank you for this observation. Indeed, sometimes it does appear that TSB1 is “stronger” than TSB2, as is evident in the lane 3 versus lane 4 of Fig. 3C, although this is not evident in Fig. 3B, or indeed, when the TSB cocktail was compared side-by-side with individual TSB1 or TSB2 applied separately (Appendix Fig. S7). We believe the occasional variance to be more likely due to variability within an experiment, as the stronger impact of TSB1 was not consistently observed.

C27: *Fig.3D - the in vivo effect of TBS1+2 is very limited. I do not agree with the authors that there is a substantial protection (page 7). There is for example no effect on the MLI (table S4).*

R27: Thank you for this comment. The target-site blockers resulted in a 25% increase in alveolar number, which is not substantial; however, the septal thickness was normalised, which is substantial. We have, however, toned down our assessment, and removed mention of a “substantial protection”. The point about the MLI is well-taken. The explanation relates to the stereology approach, where the MLI represents the “average” MLI assessed over the entire lung, and using the stereological approach, is unfortunately impacted by changes in the septal thickness. For this reason, the MLI is not a primary readout in stereological studies, and thus not presented in Fig. 1 to Fig. 4.

C28: *How was asma and pdgfra + cells quantified? Please, add a representative image as supplemental.*

R28: Thank you for this comment. Every step of the flow cytometry protocol for the determination of PDGFR α cells is now presented, in Appendix Fig. S13. Additionally, original representative flow cytometry lots for α SMA are provided in Appendix Fig. S8, S10, and S11.

C29: *Table S3 - Is the genotype of mice to the right in the table written wrong? If the mice were treated with tamoxifen they should be denoted miR-34ai(delta)PC/i(delta)PC and not mir-34a fl/fl, right?*

R29: Thank you for noticing this error, which has now been corrected.

C30: *Table S5 - how come the MLI does not improve after antimir34a during hyperoxia? Even though there is no significant difference, the trend instead suggests that the MLI gets worse. Please, comment.*

R30: Thank you for this question. As mentioned in the response to C27, in the stereology approach to the analysis of lung structure, the MLI represents the “average” MLI assessed over the entire lung, and using the stereological approach, is unfortunately impacted by changes in the septal thickness. For this reason, the MLI is not a primary readout in stereological studies, and thus not presented in Fig. 1 to Fig. 4.

C31: *Page 17 - explain/add reference to G*Power 3.1.9.2.*

R31: Thank you for this question. This reference has now been added (Faul *et al.*, 2018).

C32: *There is a protocol for primary lung fibroblasts and culture, where were these primary cell cultures used? The results mentioned seems to all come from the cell line MLg.*

R32: Thank you for this question. We have now include some data using primary mouse lung fibroblasts (Appendix Fig. S19). Therefore, the protocol has now been retained.

I hope that the reviewers will find that we have satisfactorily addressed all of their concerns.

Thank you for the submission of your revised manuscript to EMBO Molecular Medicine. We have now received the enclosed reports from the referees who were asked to re-assess it. As you will see the reviewers are now supportive, and I am pleased to inform you that we will be able to accept your manuscript pending the following minor editorial amendments.

***** Reviewer's comments *****

Referee #3 (Comments on Novelty/Model System for Author):

As stated in my previous review, the authors have gone to great lengths to use multiple systems to test their hypotheses. They have adequately addressed my questions and I believe that the manuscript is now ready for publication.

Referee #4 (Comments on Novelty/Model System for Author):

I am satisfied with the authors' replies, both to mine and other reviewers' questions and comments.